# A computational method for predicting the most likely evolutionary trajectories in the stepwise accumulation of resistance mutations

**Ruth Charlotte Eccleston[1]\*, Emilia Manko[1], Susana Campino[1], Taane G Clark[1,2], Nicholas Furnham[1]**

[1]Department of Infection Biology, London School of Hygiene and Tropical Medicine, London, United Kingdom; [2]Department of Infectious Disease Epidemiology, London School of Hygiene and Tropical Medicine, London, United Kingdom

**Abstract** Pathogen evolution of drug resistance often occurs in a stepwise manner via the accumulation of multiple mutations that in combination have a non-additive impact on fitness, a phenomenon known as epistasis. The evolution of resistance via the accumulation of point mutations in the DHFR genes of *Plasmodium falciparum* (*Pf*) and *Plasmodium vivax* (*Pv*) has been studied extensively and multiple studies have shown epistatic interactions between these mutations determine the accessible evolutionary trajectories to highly resistant multiple mutations. Here, we simulated these evolutionary trajectories using a model of molecular evolution, parameterised using Rosetta Flex ddG predictions, where selection acts to reduce the target-drug binding affinity. We observe strong agreement with pathways determined using experimentally measured IC50 values of pyrimethamine binding, which suggests binding affinity is strongly predictive of resistance and epistasis in binding affinity strongly influences the order of fixation of resistance mutations. We also infer pathways directly from the frequency of mutations found in isolate data, and observe remarkable agreement with the most likely pathways predicted by our mechanistic model, as well as those determined experimentally. This suggests mutation frequency data can be used to intuitively infer evolutionary pathways, provided sufficient sampling of the population.

**\*For correspondence:**
charlotte.eccleston@lshtm.ac.uk

**Competing interest:** The authors declare that no competing interests exist.

## Editor's evaluation

Predicting the evolutionary path towards resistance through successive mutations is an important problem. This valuable study reports on the evolution of resistance to antifolates using computational predictions of changes in drug binding affinity. The findings generally rely on solid analyses although some of the claims are only partially supported because the computational predictions on the effects of mutations are only partially validated by prior experimental data. This study will be of interest to microbiologists interested in the evolution of drug resistance.

## Introduction

The development of new antimicrobial therapeutics and the design of successful drug deployment strategies to reduce the prevalence of resistance require an understanding of the underlying molecular evolution. Antimicrobial resistance (AMR) poses a huge global health threat through a wide range of mechanisms (*Sun et al., 2019*; *Davies and Davies, 2010*; *Levy and Marshall, 2004*; *Rodrigues et al., 2016*). One of the major routes to resistance, and focus of this work, is genomic variation

within protein coding regions. Of particular significance are single-nucleotide polymorphisms (SNPs) in the antimicrobial target gene that alter the protein structure and prevent efficient binding of the antimicrobial drug. Provided these SNPs do not prevent the target from carrying out its function, the resistant strains will proliferate within the population (*Blair et al., 2015*).

The evolution of resistance is affected by the interplay between selection for resistance, selection for protein function, drug concentration, and mutational bias, and it is also influenced by a phenomenon known as epistasis (*Weinreich et al., 2006*; *Lozovsky et al., 2009*; *Jiang et al., 2013*).

Epistasis between mutations within the same protein arises due to energetic interactions between the amino acids, where the impact of a mutation depends upon the protein sequence (*Starr and Thornton, 2016*). When epistasis occurs between two or more mutations, their combined impact on protein fitness or a physical trait, such as stability or binding affinity, does not equal the sum of their independent impacts.

Epistasis can arise either as a result of direct or indirect interactions between residues resulting in non-additivity in a physical property, such as stability and affinity (non-trivial epistasis), or as a result of a nonlinear mapping from sequence to protein function or fitness (trivial epistasis). Trivial epistasis, also known as nonspecific epistasis, effects a much larger number of mutations than non-trivial epistasis, because all mutations that impact a physical property that maps nonlinearly to function or fitness will interact epistaically with one another. Trivial epistasis has been widely studied and observed for mutations that are additive with respect to physical properties including stability, ligand binding affinity, function, and folding (*Gong et al., 2013*; *Bloom et al., 2010*; *McKeown et al., 2014*; *Lunzer et al., 2005*).

Non-trivial (or specific) epistasis impacts a smaller number of mutations as it requires direct or indirect interactions between amino acid side chains (*Dickinson et al., 2013*; *Pantoliano et al., 1989*; *Olson et al., 2014*) or ligands (*Adams et al., 2019*; *Anderson et al., 2015*) or a dependence of one mutation on a structural change caused by another (*Ortlund et al., 2007*; *Dellus-Gur et al., 2015*). Non-trivial epistasis is therefore associated with stronger evolutionary constraints, historical contingency, and lower reversibility of substitutions in evolutionary trajectories. Here, we are explicitly considering mutations interacting via non-trivial epistasis, where their interactions result in non-additivity in drug binding affinity. However, our fitness function maps binding affinity to fitness nonlinearly, and so epistasis may also arise at the fitness level in our simulations of evolutionary trajectories, but we do not investigate this further.

Epistasis determines the order of fixation of mutations and the accessibility of evolutionary trajectories to resistance phenotypes (*Weinreich et al., 2006*; *Weinreich et al., 2005*) and has been observed in the evolution of many pathogens (*Khan et al., 2011*; *Gong et al., 2013*; *Sanjuán et al., 2005*), including the evolution of resistance in *Plasmodium falciparum* (*Lozovsky et al., 2009*; *Sirawaraporn et al., 1997*) and *Plasmodium vivax* (*Jiang et al., 2013*). It may also have important consequences for the success of AMR management strategies that aim to reduce resistance via the cessation of use of a particular drug, which theoretically should result in reversion of resistance mutations, due to the fitness cost incurred in the absence of the drug (*Melnyk et al., 2015*; *Vogwill and MacLean, 2015*). However, the success of this strategy has been mixed, and in some cases bacterial populations remained resistant (*Costelloe et al., 2010*; *Enne, 2010*; *Sundqvist et al., 2010*), likely due to compensatory mutations (a type of epistasis), which mitigate the deleterious impact of resistance mutations, allowing them to remain in a population and thus retain resistance even in the absence of drug selection pressures (*Andersson and Hughes, 2011*).

Fragment-based drug discovery and AMR surveillance strategies require methods to predict evolutionary trajectories to resistance. For example, by identifying mutations involved in resistance trajectories that reduce the effectiveness of an antimicrobial drug, specific regions of a target molecule can be exploited or avoided, thus creating 'evolution proof' drugs. Therefore, understanding how epistasis arises and predicting which mutations will interact is important for anticipating future mutations, designing new drugs and developing strategies to minimise resistance. Indeed, *Zhang et al., 2021*, applied an integrated computational and experimental approach and identified a novel compound that inhibits both wild-type and trimethoprin-resistant *Escherichia coli* DHFR. This is a promising strategy to prevent or delay the evolution of resistance using knowledge of resistant variants and developing drugs to target them and the wild-type simultaneously.

Evolution towards drug-resistant phenotypes in malaria species *P. falciparum* and *P. vivax* has been shown to occur in a stepwise manner, due to epistatic interactions between mutations, and the most likely trajectories to resistance phenotypes have been predicted using experimental measures of resistance (*Lozovsky et al., 2009*; *Jiang et al., 2013*; *Sirawaraporn et al., 1997*).

*P. falciparum* and *P. vivax* parasites cause the majority of malaria infections and have evolved strong resistance to many antimalarial drugs, including pyrimethamine (*Sirawaraporn et al., 1997*) and sulfadoxine (*Wang et al., 1997*). There were an estimated 241 million new cases of malaria worldwide in 2020, resulting in approximately 627,000 deaths predominately among children under 5 years of age (*WHO, 2021*). *P. falciparum* malaria has been treated with the combination drug sulfadoxine-pyrimethamine (SP) since 1970s, which targets the folate metabolic pathway. Numerous resistance mutations have arisen within its genome as a result of SNPs in *P. falciparum* dihydrofolate reductase (*Pf*DHFR) and dihydropteroate synthase (*Pf*DHPS) genes, which are the targets of pyrimethamine and sulfadoxine, respectively (*Wang et al., 1997*; *Brooks et al., 1994*). Although SP is not usually used to treat *P. vivax*, co-infections with *P. falciparum* have meant SP resistance mutations have also arisen in the *P. vivax* genome (*Snounou and White, 2004*). The enzymes of the folate pathway are largely conserved across *Plasmodium* species, and so polymorphisms in equivalent positions have been observed in *P. vivax* DHFR (*Pv*DHFR) and DHPS (*Pv*DHPS) and are thought to confer resistance to SP (*Korsinczky et al., 2004*; *Hastings et al., 2004*).

The DHFR gene encodes an enzyme that uses NADPH to synthesize tetrahydrofolate, a co-factor in the synthesis of amino acids (*Kompis et al., 2005*) and pyrimethamine acts to disrupt this process, thereby blocking DNA synthesis and slowing down growth. Stepwise acquisition of multiple mutations leading to resistance to pyrimethamine has been observed in both *Pf*DHFR (*Lozovsky et al., 2009*; *Sirawaraporn et al., 1997*) and *Pv*DHFR (*Jiang et al., 2013*).

(Note on notation: lists of single mutations are written *X, Y, Z*, multiple mutations are written *X,Y,Z* (i.e. no space between the commas and the mutations) and pathways are written *X/Y/Z* to denote the order of fixation).

Resistance in *Pf*DHFR has been studied extensively and a combination of four mutations – Asn-51 to Ile (N51I), Cys-59 to Arg (C59R), Ser-108 to Asn (S108N), and Ile-164 to Leu (I164L) – has been reported to result in resistance to pyrimethamine (*Ferlan et al., 2001*) by altering the binding pocket and reducing the affinity for the drug (*Yuthavong et al., 1999*). Epistasis in both pyrimethamine binding free energy and the concentration required to inhibit cell growth by 50% (IC50) has been observed experimentally for combinations of these four mutations (*Lozovsky et al., 2009*; *Sirawaraporn et al., 1997*). This means that mutations which on their own are not associated with a resistance phenotype, can be when in combination with other mutations. Epistasis between these mutations has been shown to determine the evolutionary trajectories to the quadruple mutation N51I,C59R,S108N,I164L, which is strongly associated with pyrimethamine resistance (*Lozovsky et al., 2009*).

A similar investigation was conducted into the homologous set of *Pv*DHFR mutations – Asn-50 to Ile (N50I), Ser-59 to Arg (S58R), Ser-117 to Asn (S117N), and Ile-173 to Leu (I173L) – and the accessible evolutionary trajectories to the quadruple mutation (*Jiang et al., 2013*), some combinations of which have been observed to result in pyrimethamine resistance both in vivo and in vitro (*Hastings et al., 2004*; *Hawkins et al., 2007*). Evolutionary simulations accounting for growth rates, IC50 measurements for increasing concentrations of pyrimethamine and nucleotide bias predicted the most likely pathways to the quadruple mutation for different drug concentrations. The observed trajectories at each concentration were influenced by epistasis between the mutations and the adaptive conflict between endogenous function and acquisition of drug resistance. These studies, along with other investigations (*Weinreich et al., 2006*; *Tamer et al., 2019*), have highlighted the prevalence of epistasis among resistance mutations and the importance of considering epistatic interactions between mutations when predicting evolutionary trajectories to drug resistance.

The epistasis between the four *Pf*DHFR mutations and the four *Pv*DHFR mutations may prevent reverse evolution from resistant to susceptible phenotypes, with S108N and S117N acting as pivot point mutations creating an epistatic ratchet against reverse evolution towards the wild-type and suggesting that the removal of pyrimethamine from use will not result in a reduction of resistance (*Ogbunugafor et al., 2016*; *Ogbunugafor and Hartl, 2016*; *Ogbunugafor, 2022*).

The predictability of evolution is a central topic in biology of interest to experimentalists and theorists alike (*Achaz et al., 2014*; *Lobkovsky and Koonin, 2012*; *Szendro et al., 2013*) (for a review of

the topic, see *de Visser and Krug, 2014*). By using experimentally measured values to characterise the empirical fitness landscapes and simulate evolutionary trajectories, the work in *Lozovsky et al., 2009*, and *Jiang et al., 2013*, is determining the predictability of evolution in these landscapes by assessing which trajectories are accessible and the level of determinism associated with the evolution. Whilst such experimental methods have been successful in capturing epistasis, characterising evolutionary landscapes, and predicting evolutionary trajectories, they are expensive and time consuming.

The development of computational methods to predict resistance trajectories would enable fast and efficient predictions and would be more widely accessible than lab-based methods. Computational tools could help narrow down the pool of mutations to be studied experimentally and would also be applicable to targets that are difficult to study. Some target-specific computational tools to predict individual resistance mutations have been developed (*Karmakar et al., 2020*; *Portelli et al., 2020*). However, such tools are target specific and so not generalisable. Furthermore, they only consider independent mutations on a single structure and so ignore epistasis between resistance mutations. Therefore, they are not suitable for predicting evolutionary trajectories to resistance.

To determine a generalisable computational method to predict evolutionary trajectories to resistance, we need to consider the main determinants of resistance. Previous studies have had success predicting the evolutionary escape of norovirus escaping a neutralising antibody using a biophysical fitness model based on capsid folding stability and antibody binding affinity (*Rotem et al., 2018*). *Rodrigues et al., 2016*, investigated three mutations in *E. coli* DHFR associated with trimethoprim resistance and considered activity, binding affinity, fold stability, and protein abundance. They found that whilst resistance is a balance between these factors, when changes in protein abundance are small, binding affinity is the single most predictive trait of resistance, especially at later points in evolution. Therefore, we decided to investigate if predictions of binding affinity change can be used to predict the order of fixation of resistance mutations involved in evolutionary trajectories to resistance.

Rosetta Flex ddG (*Barlow et al., 2018*) is the current state-of-the art method for predicting changes in protein-protein and protein-ligand binding free energy. Rosetta is a software suite for macromolecular modelling and design that uses all-atom mixed physics- and knowledge-based potentials, and provides a diverse set of protocols to perform specific tasks, such as structure prediction, molecular docking, and homology modelling (*Alford et al., 2017*). The Flex ddG protocol has been found to perform better than machine learning methods and comparably to molecular dynamics methods when tested on a large dataset of ligand binding free energy changes upon protein mutation (*Aldeghi et al., 2018*; *Aldeghi et al., 2019*). However, its ability to capture epistasis has not yet been tested. Therefore, we investigated how well Flex ddG can capture epistasis between resistance mutations in *Pf*DHFR and observed a good agreement with experimental data.

Next, we used the Flex ddG predictions to parameterise a fitness function applied in an existing model of molecular evolution. We used this method to predict evolutionary trajectories to known resistant quadruple mutants in both *Pf*DHFR and *Pv*DHFR, where the evolutionary trajectories have been studied experimentally (*Lozovsky et al., 2009*; *Jiang et al., 2013*). Good agreement was observed between the most likely trajectories to the quadruple mutations predicted by our model and those predicted experimentally. This suggests that whilst resistance is a trade-off between many biophysical parameters, a simple model considering only binding affinity changes may be effective at predicting resistance trajectories.

The main advantage of this approach is that it does not require access to an experimental 'wet' lab and can be carried out by anyone with access to a high-performance computer. It is generalisable to any antimicrobial drug that acts by binding to its target and can be easily applied to any drug-target complex for which there is an available structure. Therefore, it can be used to study complexes and systems that might be problematic experimentally. It enables accurate assessment of the predictability of the evolutionary landscape and can predict whether we would expect to see constrained evolutionary trajectories on a fitness landscape as a result of epistatic interactions in drug binding free energy.

In addition, we analysed if evolutionary pathways can be inferred from the frequency of mutations found in isolate data. We determined the frequency of mutations in *Pf*DHFR and *Pv*DHFR, and inferred the most likely evolutionary pathways under the assumption that the most likely mutation at each step corresponds to the most frequent mutation. We carried out this analysis first upon a combined set of global isolates and then upon isolates from individual regions. The most likely pathways inferred from

the global isolate data agreed remarkably well with both the experimentally determined pathways and the pathways predicted by our computational method. This suggests evolutionary trajectories can be inferred from the frequency of mutations observed in isolate data, provided adequate sampling of the population. When considering geographical regions separately, the inferred pathways from several regions agreed well with the experimental pathways and our predicted pathways, however the most likely pathways inferred in some regions differed from the main pathways, highlighting the importance of considering the evolution in different regions separately.

## Results
### Rosetta Flex ddG captures general trends in binding free energy changes and epistasis

We investigated if Flex ddG predictions agree with experimentally measured binding free energy and if these predictions can be used to calculate the non-additivity in binding free energy (interaction energy), which for a double mutant defines the epistasis between the two mutations and, for a triple mutant or higher, captures the level of epistatic interactions but does not quantify it. We calculated the interaction energy by finding the difference between the predicted change in binding free energy of a multiple mutation and the sum of the predictions of their independent binding free energy changes. A positive value of the interaction energy indicates the sum of the independent impacts is more destabilising than the impact of the multiple mutation and a negative value indicates the sum is less destabilising than the combined impact.

The change in binding free energy was predicted using Flex ddG for the combinatorially complete set of the four *Pf*DHFR pyrimethamine resistance mutations N51I, C59R, S108N, and I164L.

We compared the predictions to the data from *Sirawaraporn et al., 1997*, in which they determined binding free energy changes for a subset of the possible combinations of mutations, the sum of the independent mutations (calculated for multiple mutants to compare to the experimentally

**Table 1.** Correlation between Flex ddG predictions for 250 runs and experimental data (see table 4 of *Sirawaraporn et al., 1997*) for *P. falciparum* dihydrofolate reductase (*Pf*DHFR) pyrimethamine resistance mutations.

| Mutation | $\Delta\Delta G_{exp}$*(kcal/mol) | Exp. sum† | Exp I.E.‡ | $\Delta\Delta G_{FlexddG}$ §(kcal/mol) | Sum¶ | I.E.§** |
|---|---|---|---|---|---|---|
| N51I | –0.783 | | | –0.124 | | |
| C59R | –0.184 | | | –0.033 | | |
| S108N | 1.297 | | | 0.312 | | |
| I164L | –0.351 | | | –0.323 | | |
| N51I,S108N | 1.89 | 0.514 | 1.376 | –0.166 | 0.188 | –0.354 |
| C59R,S108N | 2.29 | 1.113 | 1.177 | 0.399 | 0.279 | 0.119 |
| N51I,C59R,S108N | 2.595 | 0.33 | 2.265 | 0.162 | 0.155 | 0.007 |
| C59R,S108N,I164L | 3.283 | 0.762 | 2.521 | 0.018 | –0.043 | 0.061 |
| N51I,C59R,S108N,I164L | 3.761 | –0.021 | 3.782 | 0.301 | –0.168 | 0.469 |
| Pearson correlation | | | | 0.611 | 0.660 | 0.756 |
| Correctly classified | | | | 8/9 | 4/5 | 4/5 |

*Experimentally measured *Pf*DHFR pyrimethamine binding free energy change data from *Sirawaraporn et al., 1997*.

†Sum of experimental values of binding free energy change for independent mutations.

‡Interaction energy calculated as the difference between experimentally measured values of binding free energy change of multiple mutant compared to the sum of the independent mutations involved.

§Change in *Pf*DHFR pyrimethamine binding free energy predicted by Flex ddG calculated as the average of the distribution of runs. Free energy predictions from Rosetta are in Rosetta Energy Units, however the authors of Flex ddG applied a generalised additive model to reweight the predictions and make the output more comparable to units of kcal/mol (*Barlow et al., 2018*).

¶Sum of Flex ddG predictions for independent mutations.

** Interaction energy calculated as the difference between Flex ddG predicted binding free energy change of multiple mutant compared to the sum of the independent mutations.

determined binding free energy changes of multiple mutants), and the interaction energy of the multiple mutants (*Table 1*). A positive $\Delta\Delta G$ value indicates a destabilising mutation and a negative $\Delta\Delta G$ value indicates a stabilising mutation. (Note: Rosetta Flex ddG calculates the change in binding free energy as $\Delta\Delta G = \Delta G_{mut} - \Delta G_{WT}$, whereas *Sirawaraporn et al., 1997*, calculated the change as the reverse, $\Delta\Delta G = \Delta G_{WT} - \Delta G_{mut}$, where *WT* indicates the wild-type free energy and *mut* indicates the mutant free energy. Therefore, in *Sirawaraporn et al., 1997*, a mutation that destabilised the binding corresponded to a negative $\Delta\Delta G$, whilst here we have reversed the signs of their data to enable comparison with our predictions.)

The authors of the Flex ddG protocol suggest conducting a minimum of 35 runs and taking the average of the distribution as the prediction for that mutation (*Barlow et al., 2018*). We found the average of the distributions converges and the rank order of the mutations is constant at around 250 runs (*Appendix 1—figure 3* and *Appendix 1—figure 4*). We compared the predictions for 250 runs and the data from *Sirawaraporn et al., 1997* (*Table 1*) and observed a correlation of 0.611 for the binding free energy data (with a p-value of 0.04 for a one-sided t-test), 0.660 for the sum of the independent predictions for multiple mutants, and 0.756 for the interaction energy. We found 8/9 binding free energy predictions were correctly classified, 4/5 of the sum of the independent predictions were correctly classified, and 4/5 of the interaction energies were correctly classified. We used a confusion matrix to determine how accurate Flex ddG is at classifying mutations as stabilising or destabilising, which gave an accuracy of 0.89, a sensitivity of 0.83, and a specificity of 1.0, suggesting Flex ddG performs well overall. The p-value associated with the confusion matrix was 0.14, despite the high accuracy. This is likely due to the small sample size making it difficult to determine significance.

Comparing the predictions for 35 runs (*Appendix 1—figure 1*, *Appendix 1—table 1*) and 250 (*Appendix 1—figure 2*, *Table 1*) runs, 250 runs provides the best trade-off between accuracy and efficiency (see Appendix 1 for detailed discussion). Therefore, we will be discussing the predictions for *n*=250 going forward.

Mutation S108N was the only single mutation to destabilise pyrimethamine binding in both the experimental data and the Flex ddG predictions. However, in the experimental data the double mutation N51I,S108N is more destabilising to binding than single mutation S108N, but the Flex ddG prediction was stabilising. The triple mutation C59R,S108N,I164L was found experimentally to be the most destabilising of the triple mutations, however Flex ddG predicted it to be only mildly destabilising and the least destabilising of the triple mutations. Furthermore, the quadruple mutation was found experimentally to have the most destabilising impact out of all combinations of single and multiple mutations, however, Flex ddG predicted it to be less destabilising than the double mutation C59R,S108N and single mutation S108N.

Considering the interaction energy, the incorrectly classified mutation was again N51I,S108N which was predicted to be positive, but found experimentally to be negative, because the sum of the individual predictions was destabilising but the double mutation itself was predicted to be stabilising. Both the experimental data and our predictions found that the quadruple mutation had the largest magnitude interaction energy reflecting the greatest difference between the stabilising impact of the sum of the individual mutations and the destabilising impact of the quadruple mutation itself.

We also observed large negative interaction energy between S108N and C59R, where C59R is stabilising in the wild-type background but destabilising in the background of S108N, an example of sign epistasis and in agreement with the observations of both *Sirawaraporn et al., 1997*, and *Lozovsky et al., 2009*. However, whilst the interaction energy of the triple mutation N51I,C59R,S108N was positive for both the experimental data and predictions, in our predictions its magnitude was much smaller compared to the data. Both single mutations N51I and C59R were predicted to be only marginally stabilising – almost neutral – to pyrimethamine binding, whilst in the experimental data both mutations have a large stabilising impact. Furthermore, the triple mutation was predicted to be only marginally more destabilising than single mutation S108N, resulting in the small negative interaction energy.

We conclude that although there are some disagreements between the predictions and the data, Flex ddG is able to capture the general trend of the data. However, if we use the average of the distributions as a summary metric of the predictions for the combinatorially complete set of the four mutations and try to infer a pathway through to the quadruple mutation, under the criteria that each subsequent mutation must destabilise pyrimethamine binding more than the last, then we are

unable to find a pathway through (see *Appendix 2—figure 1* for fitness hypercube). However, since the predictions capture the general trend observed in the data, and the summary metric does not fully characterise the entire distribution of predictions, we used the distributions to parameterise an evolutionary model to determine if we can predict a pathway through to the quadruple mutation and if the predicted evolutionary trajectories agree with experimentally determined evolutionary trajectories.

## A thermodynamic evolutionary model predicts the most likely evolutionary trajectories to quadruple mutations in both *Pf*DHFR and *Pv*DHFR

Whilst the fitness landscape of DHFR is a function of many factors including stability, abundance, and activity (*Bershtein et al., 2015*; *Rodrigues et al., 2016*), we sought to construct a simple biophysical model that could be parameterised using existing computational tools. Therefore, we used a simple model that considered only predicted changes in binding free energy, that would be easy to implement, undertaken purely computationally with good accuracy and without the need for extensive wet lab experiments. We sought to determine if simulated trajectories using this simple fitness model could capture observed evolutionary trajectories, despite not considering all factors that contribute to DHFR fitness. We simulated the evolutionary trajectories to the quadruple mutants described above for the genes *Pf*DHFR and *Pv*DHFR using an evolutionary model, adapted from previous studies (*Eccleston et al., 2021*; *Pollock et al., 2017*; *Pollock et al., 2012*). In this model, selection acts to reduce the binding affinity between target protein and the antimalarial drug with which the mutations have been associated with resistance. Briefly, starting from the wild-type protein, we randomly sample a value from the Flex ddG distributions for each of the four single mutations and calculate the fitness of the mutated protein (*Equation 1*) and the fixation probability (*Equation 2*). A mutation is then chosen with a probability proportional to the fixation probability and this is repeated until the quadruple mutation is reached. If the set of sampled mutations at a step all have a fixation probability of zero, the algorithm terminates at that point in the pathway and begins the next run at the single mutation step. Therefore, it is not guaranteed that a run will reach the quadruple mutation. We carried out 50,000 runs and determined (i) the number of runs that reached a single, double, triple, or the quadruple mutation before the run ended (files ending '_endpoint_numbers.csv' on Zenodo), (ii) the frequency of the observed trajectories up to the quadruple mutation, including trajectories that terminated before the quadruple mutation (files ending '_pathway_endpoints.csv'), (iii) the frequency at which a mutational step was chosen in all runs (files ending '_total_pathway_probabilities.csv'), and (iv) the most likely trajectories to the quadruple mutation predicted by our simulations (files ending '_quadruple_pathways.csv').

To determine how well our simulations reflect the evolutionary process to the *Pf*DHFR quadruple mutation N51I,C59R,S108N,I164L, we compared our results to experimentally determined evolutionary trajectories as presented by *Lozovsky et al., 2009*. In our simulations, the quadruple mutation was reached in approximately 8% of runs (see supplementary file 'PfDHFR_endpoint_numbers.csv' on Zenodo). The majority of the runs (66%) terminated at a double mutation, with S108N/C59R the most likely trajectory over all. The algorithm was often unable to proceed passed S108N/C59R because the Flex ddG distribution for C59R,S108N is concentrated around large destabilising values (*Appendix 1—figure 2f*) whilst the distributions of the two possible next steps, N51I,C59R,S108N and C59R,S108N,I164L, are concentrated around lower destabilising values (*Appendix 1—figure 2g, h*). Therefore, in many instances, the change in binding free energy caused by the next step in the pathway was predicted to be stabilising, and thus were not be chosen by the algorithm. This demonstrates the dependence of the method upon the accuracy of the Flex ddG. In contrast, the majority of runs in the simulations based on IC50 measurements presented in *Lozovsky et al., 2009*, reached the quadruple mutation (see Figure 2 in *Lozovsky et al., 2009*).

However, since we are interested in how epistasis influences the order of fixation of mutations in an evolutionary trajectory to a high-resistance quadruple mutation, we compared the most likely trajectories to the quadruple mutation predicted by our simulations to the most likely trajectories to the quadruple mutation predicted in *Lozovsky et al., 2009*, and observed good agreement. To summarise the overall performance of the computational method, we calculated the overall intersect between the top pathways predicted by our model and those predicted by *Lozovsky et al., 2009*, and the average overlap, which calculates the average of the intersect for all depths, thus weighting

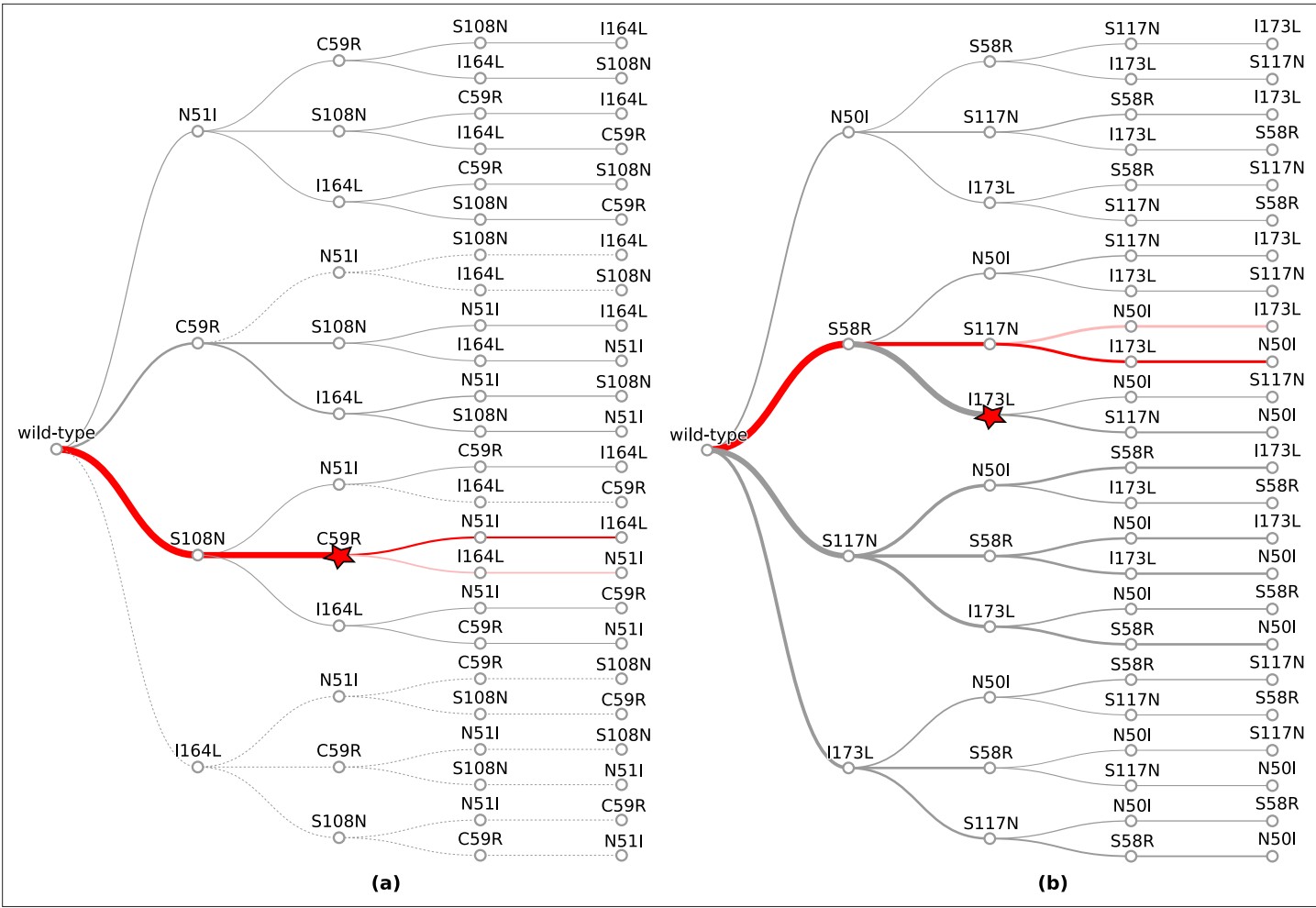

**Figure 1.** The probability of simulated evolutionary pathways to quadruple mutations (**a**) N51I,C59R,S108N,I164L in *P. falciparum* dihydrofolate reductase (*Pf*DHFR) and (**b**) N50I,S58R,S117N,I173L in *P. vivax* DHFR (*Pv*DHFR). Line thickness indicates the total probability of a mutation when considering all pathways it can occur in, determined from the frequency of that step in all realised mutational pathways from all runs. Dotted lines indicate zero probability of a mutation at that step. The most likely pathway in total is denoted by a red star. The most likely pathway to the quadruple mutation is highlighted in dark red and the second most likely pathway to the quadruple mutation is highlighted in lighter red. The probabilities corresponding to these plots can be found in supplementary files 'PfDHFR_total_pathway_probabilities.csv' and 'PvDHFR_total_pathway_probabilities. csv' on Zenodo for (**a**) and (**b**), respectively.

the top ranks higher. The intersect between the top 10 pathways and those presented in *Lozovsky et al., 2009*, is 0.6 and the average overlap is 0.687. The top two most likely trajectories predicted by our model to the *Pf*DHFR quadruple mutation were S108N/C59R/N51I/I164L and S108N/C59R/ I164L/N51I, respectively (*Figure 1a*) which correspond to the top two most likely pathways to the quadruple mutation determined in *Lozovsky et al., 2009*. The third most likely trajectory to the quadruple mutation in *Lozovsky et al., 2009*, was predicted to be S108N/N51I/C59R/I164L, however this pathway was predicted to be unlikely in our simulations, due to the fact that the distribution of Flex ddG predictions for double mutation N51I,S108N was mostly stabilising to pyrimethamine binding (*Appendix 1—figure 2e*), whereas all of the S108N distribution was destabilising to pyrimethamine binding, so this step was unlikely to be chosen by the evolutionary algorithm.

Considering the frequency at which the single mutations were chosen as the first step in all simulated pathways ('PfDHFR_total_pathway_probabilities.csv' on Zenodo), S108N was the most likely single mutation and C59R was the second most likely single mutation, in agreement with the two most likely first steps in the pathways predicted in *Lozovsky et al., 2009*. The most likely pathway to a double mutation realised in all trajectories in both our simulations and the simulations in *Lozovsky*

*et al., 2009*, is S108N/C59R. Similarly, the most likely pathway to a triple mutation realised in all our simulations and in *Lozovsky et al., 2009*, was S108N/C59R/N51I.

To simulate the evolutionary pathways for *Pv*DHFR, we also carried out predictions of binding free energy changes for the homologous set of four mutations in *Pv*DHFR (N50I, S58R, S117N, and I173L). Unfortunately, binding affinity data is not available for the mutations in *Pv*DHFR to compare to the Flex ddG predictions. However, *Jiang et al., 2013*, predicted pathways to the *Pv*DHFR quadruple mutation for four pyrimethamine concentrations using simulations informed by both drug resistance and catalytic activity, to which we can compare our simulations. In their simulations, the quadruple mutation fixed in 99.8% of runs for the highest pyrimethamine concentration, but it did not fix for the three lower concentrations. In our simulations, the quadruple mutation was reached in 39% of runs ('PvDHFR_endpoint_numbers.csv' on Zenodo), whilst 51% of runs terminated at a double mutation. The most likely endpoint overall in our simulations was S58R/I173L, which occurred in 32% of runs, however this path was not a frequent trajectory observed in the simulations in *Jiang et al., 2013*. All Flex ddG runs of double mutation S58R,I173L were predicted to be destabilising and many predicted to have a medium to large impact. However, the triple mutation S58R,S117N,I173L was predicted to have a smaller destabilising impact than S58R,S117N, making pathway S58R/I173L/N50I unlikely, whilst N50I,S58R,I173L was predicted to be stabilising to pyrimethamine in all Flex ddG runs and therefore pathway S58R/I173L/N50I had a zero probability of occurring in the simulations. This resulted in many runs terminating at step S58R/I173L.

Considering the order of fixation up to the quadruple mutation, we compared the most likely evolutionary trajectories to the quadruple mutation predicted by our simulations to the most likely evolutionary trajectories to the quadruple mutation presented in *Jiang et al., 2013*, and observed good agreement for the largest of the four pyrimethamine concentrations they considered. Overall, the intersect between the top four pathways predicted by our model and the top four pathways present in *Jiang et al., 2013*, for the highest concentration of pyrimethamine is 0.75 and the average overlap is 0.604. The most likely pathway to the quadruple mutation predicted by our simulations was S58R/S117N/I173L/N50I (*Figure 1b*), which corresponds to the second most likely pathway to the quadruple mutation predicted in *Jiang et al., 2013*, for the highest pyrimethamine concentration. Our second most likely pathway to the quadruple mutation (S58R/S117N/N50I/I173L) corresponds to the first most likely pathway predicted in *Jiang et al., 2013*, for the highest pyrimethamine concentration. There were two other possible pathways to the quadruple at the highest concentration, S117N/N50I/S58R/I173L and N50I/S117N/S58R/I173L, which correspond to the fourth and twelfth most likely pathways to the quadruple in our simulations.

The first step in the evolutionary trajectories determined in *Jiang et al., 2013*, for the highest concentration was S58R whereas for the three lower concentrations it was S117N. The most likely first step in all pathways predicted by our simulations was S58R (*Figure 1b*), whilst S117N was the second most likely first step ('PvDHFR_total_pathway_probabilities.csv' on Zenodo).

To quantify the predictability of an evolutionary landscape, previous studies have calculated the Gibbs-Shannon entropy distribution of the path weights (*Szendro et al., 2013*; *de Visser and Krug, 2014*), namely $S = -\sum P_i \ln P_i$, where $P_i$ is the probability of the $i$th pathway and the value of $S$ ranges from 0 to $\ln n$ for $n$ equally likely pathways. The lower the value of $S$ the higher the predictability of the evolution, i.e., most of the probability is concentrated around a small number of pathways, suggesting epistasis is influential in constraining the accessible trajectories. The value of $S$ when considering the probability distribution of all realised evolutionary trajectories in the simulations was 1.19 for *Pf*DHFR and 2.82 for *Pv*DHFR (both simulations have an equal number of possible pathways because they have an equal number of mutations, so the values of $S$ are comparable and the maximum value of $S$ for both simulations is 4.16). This means the evolutionary trajectories were more constrained in the *Pf*DHFR simulations than in the *Pv*DHFR simulations and suggests that epistasis between the mutations plays a greater role in constraining the trajectories in the evolution of *Pf*DHFR resistance. Unfortunately, the probabilities of all possible pathways determined in *Lozovsky et al., 2009*, and *Jiang et al., 2013*, are not made available (the data is represented in pathway diagrams, the probabilities of a step are indicated by line thickness and only the probabilities of the most likely pathways annotated), therefore we cannot calculate the corresponding values of $S$ for these distributions for comparison.

## The frequency of mutations in isolate data can be used to infer evolutionary trajectories to multiple resistance mutations

It was noted in *Lozovsky et al., 2009*, that their most likely pathways to the *Pf*DHFR quadruple mutation were consistent with combinations of these four mutations observed in high frequencies in worldwide surveys of *P. falciparum* polymorphisms. To expand on this idea, we analysed the frequency of the combinations of mutations in *Pf*DHFR and *Pv*DHFR found in our isolate data to identify if there is agreement between these frequencies, the experimentally determined trajectories, and our predicted trajectories and if, therefore, isolate frequency data may be used to infer evolutionary trajectories. We inferred evolutionary trajectories from the frequency data by assuming if a specific mutation was found in high frequency (and is part of the combinatorically complete set of four mutations found in the four genes) then it is likely to be part of the evolutionary trajectory towards the quadruple mutation. To infer the first step in the most likely trajectory, we considered the frequency of single mutations of the set of four mutations considered for each gene and selected the most frequent mutation. To infer subsequent steps in the trajectory, we considered the frequency of only those mutations that contain the previous mutation and another of the set of four mutations in some combination and chose the most frequent mutation at each step. We also inferred alternative pathways which from the frequency data are less likely than the main pathway, but still a possibility due to the occurrence of intermediate mutations in the isolate data. To do this, we considered each step in the most likely trajectory and identified any other high-frequency mutations that would enable alternative pathways from the double mutation onwards. If there were no alternative pathways, we began the process again but chose the second most frequent single mutation (if applicable) and built the pathway from there. In the event of multiple alternative pathways, we are unable to quantify their relative likelihoods, only that they are less likely than the most likely pathway. It is sometimes not clear which pathway is most likely. For example, for the set of mutation frequencies *A*:9, *D*:10, *AB*:20, *CD*:2, *ABC*:50, *BCD*:1, *ABCD*:75, the most likely pathway from the method stated above would be *D/C/B/A* and the alternative pathway would be *A/B/C/D*, purely because mutation *D* is more abundant than mutation *A*. However, the frequencies of the intermediate mutations in the most likely pathway are low compared to the alternative pathway. Therefore, in these situations where it is unclear which pathway is most likely, we will not refer to any one pathway as the most likely pathway and will refer to all pathways as possible trajectories.

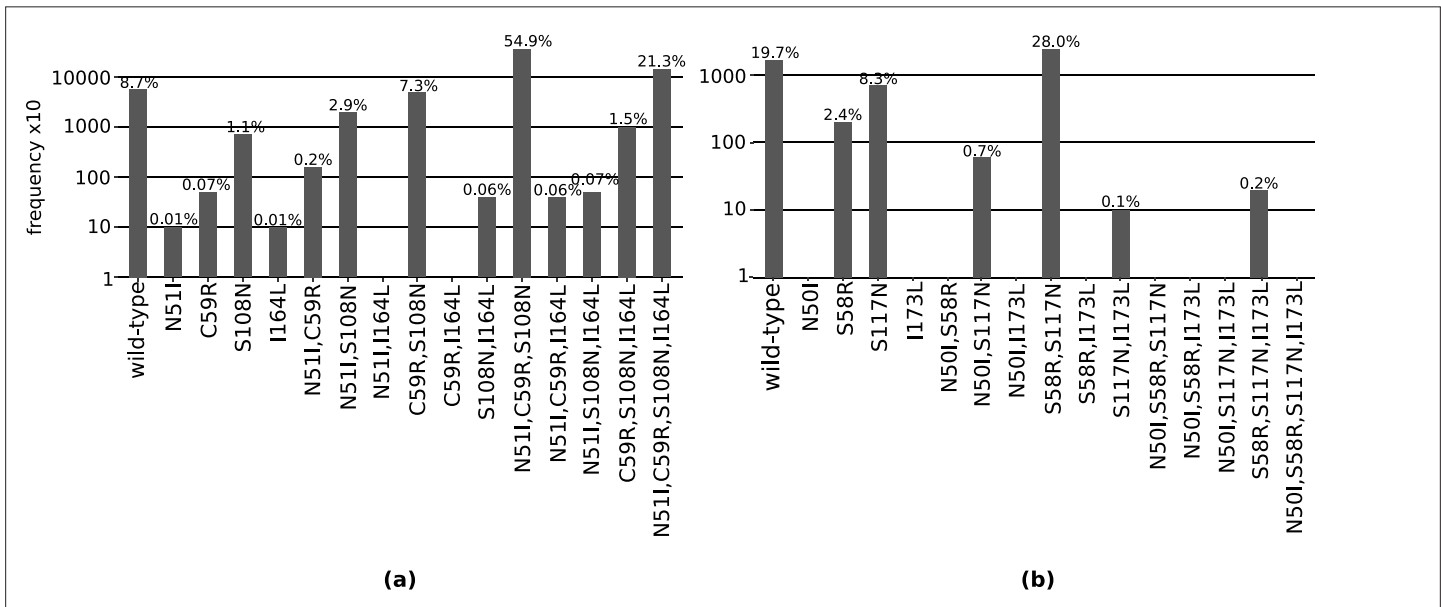

**Figure 2.** The total frequency of the combinations of mutations found in our isolate data for sets of four mutations (**a**) N51I, C59R, S108N, and I164L in *P. falciparum* dihydrofolate reductase (*Pf*DHFR), and (**b**) N50I, S58R, S117N, and I173L in *P. vivax* DHFR (*Pv*DHFR). All frequencies have been multiplied by a factor of 10 to enable clear identification of those mutations occurring in one isolate only. The frequencies are also given as the percentage of the total number of isolates, which for *Pf*DHFR is 6762 and *Pv*DHFR is 847.

Considering the total frequency of each mutation in the set of four *Pf*DHFR mutations (N51I, C59R, S108N, and I164L) in the isolate data (*Figure 2a*), S108N was the most frequent single mutation (72/6762 isolates), C59R,S108N the most frequent double mutation (496/6762 isolates), and N51I,C59R,S108N the most frequent triple mutation (3714/6762 isolates). The quadruple mutation N51I,C59R,S108N,I164L was found in 1439/6762 isolates. This suggests the pathway proceeds in the order S108N/C59R/N51I/I164L, in agreement with the most likely pathway to the quadruple mutation from both our evolutionary simulations and those using experimental data (*Lozovsky et al., 2009*).

Triple mutation C59R,S108N,I164L was found in 101/6762 isolates, suggesting that the second most likely pathway to the quadruple from our simulations and experimental data, S108N/C59R/I164L/ N51I, is a possible alternative trajectory to the quadruple mutation. Double mutation N51I,S108N was the second most frequent double mutation in the isolate data (198/6762 isolates), allowing for another alternative pathway S108N/N51I/C59R/I164L. This agrees with the third most likely pathway presented by *Lozovsky et al., 2009*, however this pathway was unlikely in our simulations.

Single mutations C59R and N51I were the second and third most prevalent single mutations in our isolate data, found in 5/6762 and 1/6762 of isolates, respectively. They were also the second and third most likely first step in our pathway predictions ('PfDHFR_total_pathway_probabilities.csv' on Zenodo). Single mutation I164L was absent from the isolate data and had zero probability of being selected as the first step of our evolutionary trajectories.

A Chi-squared analysis revealed the worldwide distribution of mutations is significantly different than would be expected if there was no preferred evolutionary pathway, and the mutations which were overrepresented were those involved in the most likely pathway inferred above S108N/C59R/N51I/ I164L (see Appendix 3 and *Appendix 3—figure 1*). This provides further support that the epistatic interactions between the mutations determine the order of fixation resulting in preferred pathways to the quadruple mutation.

Considering the set of four *Pv*DHFR mutations (N50I, S58R, S117N, and I173L), in our isolate data, the mutations S58R and S117N are fixed at these locations and the wild-type alleles are now considered to have an arginine at codon 58 and asparagine at codon 117. However in *Jiang et al., 2013*, they consider the wild-type allele to have a serine at codons 58 and 117 and therefore we have changed our definition of the wild-type allele to agree with *Jiang et al., 2013*, for ease of comparison with their evolutionary pathways and our own.

The most frequent single mutation was S117N (70/847 isolates), the most frequent double mutation was S58R,S117N (237/847), and the only observed triple mutation was S58R,S117N,I173L (2/847) (*Figure 2b*). The quadruple mutation was not observed in our isolate data, and has not been reported in the literature either. By considering the frequency of the possible combinations of mutations, we inferred the evolution towards triple mutation S58R,S117N,I173L most likely occurs via pathway S117N/S58R/I173L. This corresponds to the fifth most likely pathway to a triple mutation when considering all pathways observed in our simulations, however this pathway is not observed in any of the most frequent pathways at any of the four concentrations studied in *Jiang et al., 2013*.

Single mutation S58R was the second most frequent single mutation in our isolate data (20/847). This supports the predicted first evolutionary steps in *Jiang et al., 2013*, which were predicted to be S58R for the highest pyrimethamine concentration and S117N for three lower concentrations of pyrimethamine, which suggests both single mutations are possible, but S117N is more likely for a lower pyrimethamine concentrations. An alternative pathway to the triple mutation could therefore be S58R/S117N/I173L, which may be more likely under higher pyrimethamine concentrations. This corresponds to the most likely pathway to a triple mutation in our simulations and is part of the second most likely pathway to the quadruple mutation at the highest pyrimethamine concentration considered in *Jiang et al., 2013*.

A Chi-squared test on the frequency distributions of the single and double *Pv*DHFR mutations (the triple mutations were too infrequent to include in the analysis, see Appendix 3 and *Appendix 3— figure 1* for more details) revealed the worldwide distribution of *Pv*DHFR mutations is significantly different than would be expected if there were no preferred order of fixation of this set of mutations, with S117N and S58R,S117N being overrepresented in the single and double distributions, respectively. This supports our inference that pathway S117N/S58R/I173L is the most likely pathway in the worldwide data.

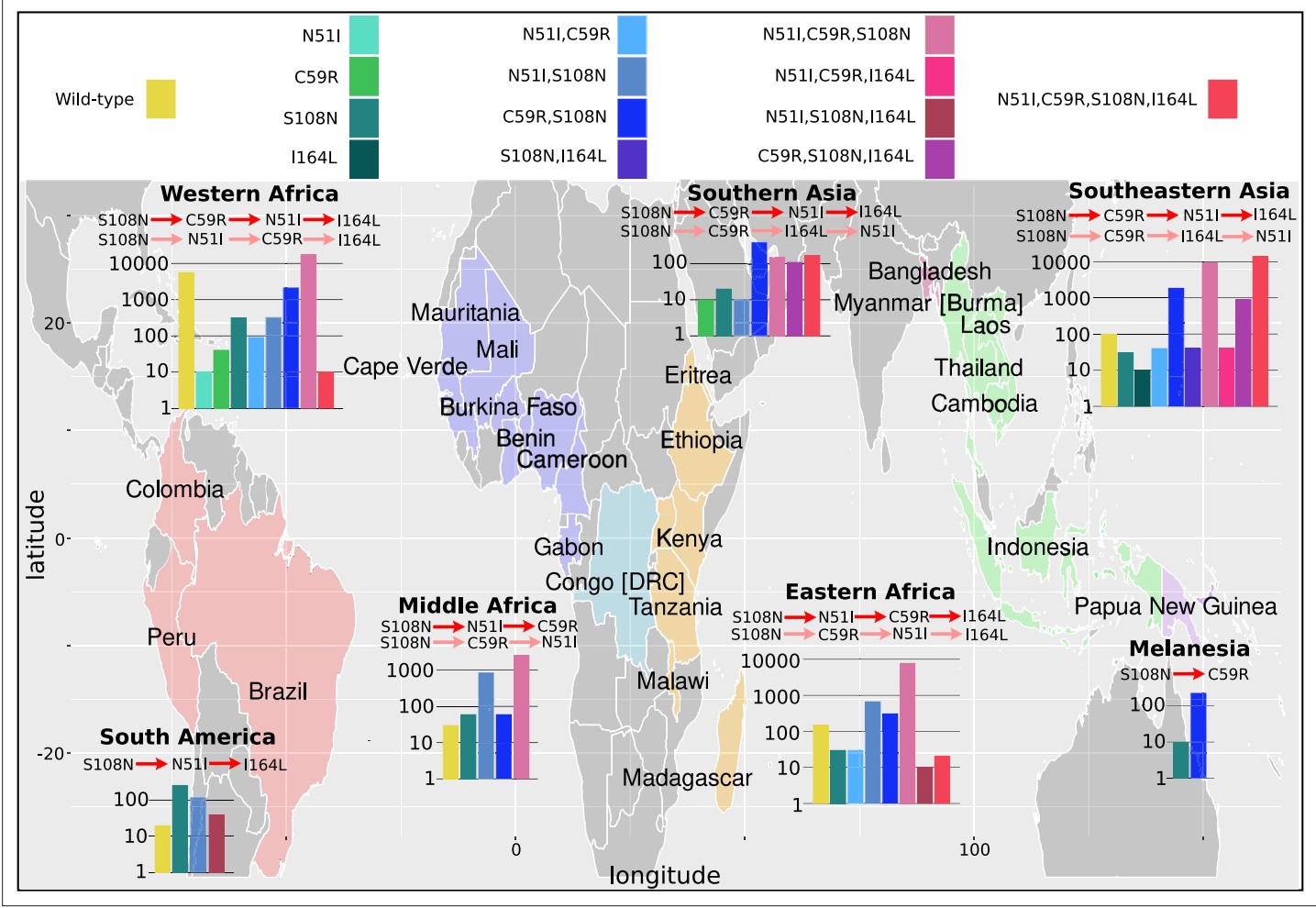

**Figure 3.** The *P. falciparum* dihydrofolate reductase (*Pf*DHFR) isolate data was grouped into seven geographical areas: South America, West Africa, Middle Africa, Eastern Africa, Southern Asia, Southeastern Asia, and Melanesia. The bar charts display the frequency (log scale) of the combinations of the four mutations N51I, C59R, S108N, and I164L. The frequency data has been multiplied by a factor of 10 to enable clear identification of those mutations occurring in one isolate only. The most likely evolutionary trajectory inferred from the frequency of combinations are included above the corresponding frequency chart from which the pathways were inferred indicated by mutations separated by dark red arrows. Alternative pathways are indicated by mutations separated by light red arrows. Where only single mutations are present a pathway is not inferred. (See Supplementary data folder 'PfDHFR/IsolateMutationFrequency' for the frequency of all mutations found in the isolate data from these regions.)

## Analysis of geographical distribution of mutations found in our isolate data reveals alternative pathways to resistance

We next considered the evolutionary trajectories by geographical location to determine if there are any differences in the inferred trajectories compared to the global trajectories, and which areas agree with the trajectories predicted by our simulations. The *P. falciparum* isolates were grouped into seven geographical regions (*Figure 3*), as defined by the United Nations Statistics Division: South America (Brazil, Colombia, and Peru), West Africa (Benin, Burkina Faso, Cameroon, Cape Verde, Cote d'Ivoire, Gabon, Gambia, Ghana, Guinea, Mali, Mauritania, Nigeria, and Senegal), Middle Africa (Congo [DRC]), Eastern Africa (Eritrea, Ethiopia, Kenya, Madagascar, Malawi, Tanzania, Uganda), Southern Asia (Bangladesh), Southeastern Asia (Cambodia, Indonesia, Laos, Myanmar, Thailand, and Vietnam), and Melanesia (Papua New Guinea). The *P. vivax* isolates were grouped into seven broad geographical regions (*Figure 4*), as defined by the United Nations Statistics Division: Central America (Mexico), South America (Brazil, Colombia, Guyana, Panama, Peru), Eastern Africa (Ethiopia, Eritrea, Madagascar, Sudan, Uganda), Southern Asia (Afghanistan, Bangladesh, India, Pakistan, Sri Lanka), Southeastern Asia (Cambodia, Indonesia, Laos, Malaysia, Myanmar, Philippines, Thailand, and Vietnam),

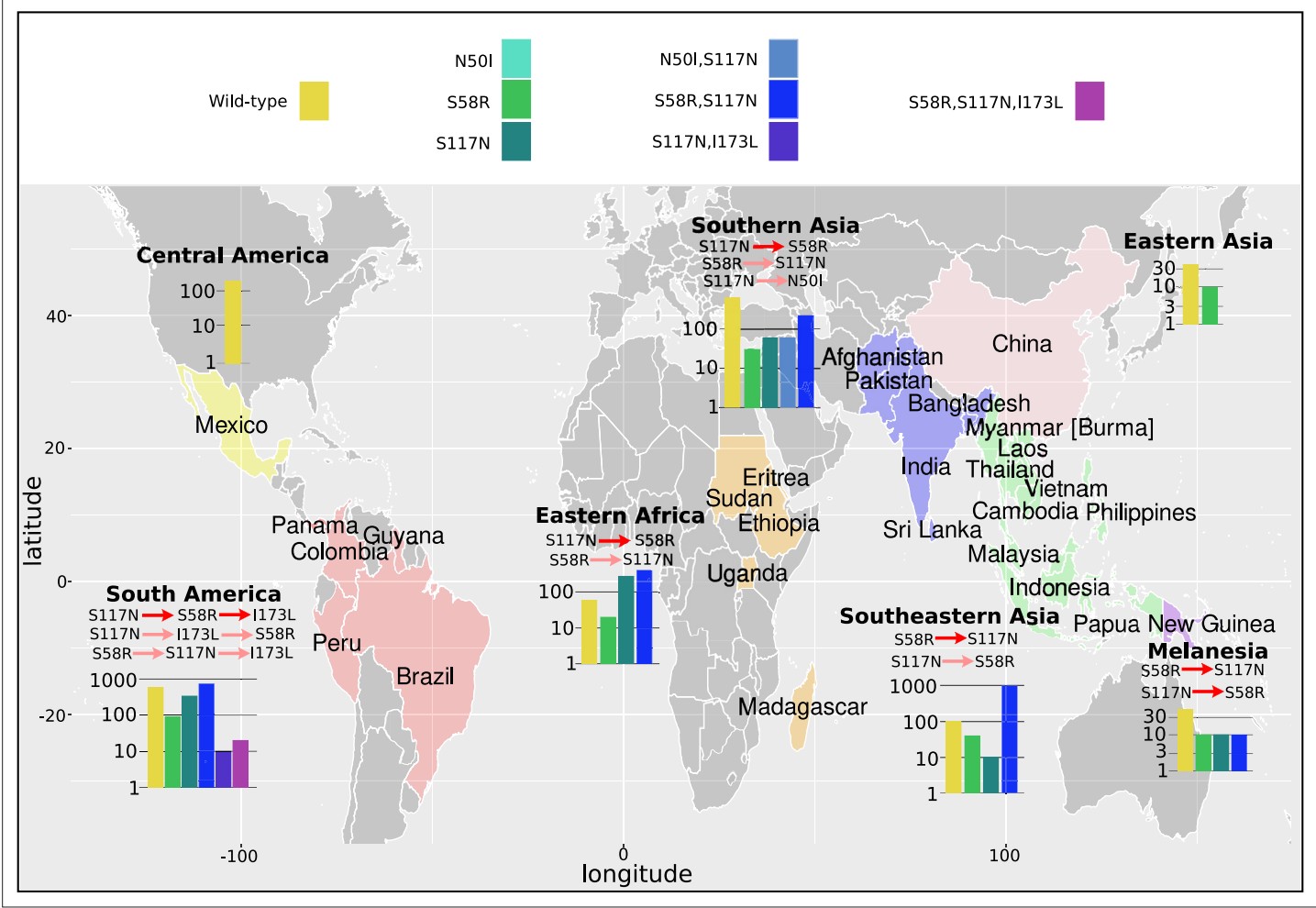

**Figure 4.** The *P. vivax* dihydrofolate reductase (*Pv*DHFR) isolate data was grouped into seven geographical areas: Central America, South America, Eastern Africa, Southern Asia, Eastern Asia, Southeastern Asia, and Melanesia. The bar charts display the frequency (log scale) of the combinations of the four mutations N50I, S58R, S117N, and I173L. The frequency data has been multiplied by a factor of 10 to enable clear identification of those mutations occurring in one isolate only. The most likely evolutionary trajectory inferred from the frequency of combinations are included above the corresponding frequency chart from which the pathways were inferred indicated by mutations separated by dark red arrows. Alternative pathways are indicated by mutations separated by light red arrows. Where only single mutations are present a pathway is not inferred.

Eastern Asia (China), and Melanesia (Papua New Guinea). The mutation frequency data discussed in this section for each country can be found in Supplementary data folders 'PfDHFR/IsolateMutation-Frequency' and 'PvDHFR/IsolateMutationFrequency'.

As in the previous section, we inferred the most likely pathway by assuming the most frequent mutation at each step corresponds to the most likely evolutionary trajectory. When inferring pathways at a regional level, it is possible we may encounter instances where genotypes with multiple mutations are observed in a specific region, but the precursor mutations in the pathway are absent. This could happen either due to insufficient sampling of the region or due to 'migration' of the variant from a neighbouring region. To infer pathways in the former case more samples would be required, whereas in the latter case we can look to the data from neighbouring regions where the variant is present and use the frequency data of the precursor mutations.

The inferred most likely pathway to the quadruple mutation agreed with the main pathway, S108N/C59R/N51I/I164L, predicted by our evolutionary model and the data presented in *Lozovsky et al., 2009*, as well as the most likely pathway inferred by considering the frequency of the worldwide *Pf*DHFR isolate data in Western Africa (S108N: 31/2594; C59R,S108N: 211/2594; N51I,C59R,S108N: 1739/2594; N51I,C59R,S108N,I164L: 1/2594), Southern Asia (S108N: 2/86; C59R,S108N: 39/86; N51I,C59R,S108N: 15/86; N51I,C59R,S108N,I164L: 17/86), and Southeastern Asia (S108N: 3/2650;

C59R,S108N: 186/2650; N51I,C59R,S108N: 920/2650; N51I,C59R,S108N,I164L: 1419/2650). Additionally, the alternative worldwide pathway S108N/C59R/I164L/N51I was inferred to be an alternative in Southern Asia (C59R,S108N,I164L: 11/86) and Southeastern Asia (C59R,S108N,I164L: 90/2650), corresponding to the second most likely pathway to the quadruple predicted by our simulations, the data in *Lozovsky et al., 2009*, and worldwide frequency data.

This main pathway to the quadruple mutation was also inferred to be a possible alternative pathway in Eastern Africa (S108N: 3/904; C59R,S108N: 31/904; N51I,C59R,S108N: 782/904, N51I,C59R,S108N,I164L: 2/904). However, in Eastern Africa S108N/N51I/C59R/I164L was the most likely pathway to the quadruple mutation (S108N,N51I: 67/904), which corresponds to the third most likely pathway presented in *Lozovsky et al., 2009*, and inferred from the total frequency data, but which was unlikely in our simulations. This was also an alternative pathway in Western Africa (S108N,N51I: 32/2594).

Furthermore, the quadruple mutation was not observed in the isolate data from Middle Africa, South America, and Melanesia. In Middle Africa, evolution up to the triple mutation N51I,C59R,S108N was observed and the most likely inferred pathway to this mutation was S108N/N51I/C59R (S108N: 6/359; N51I,S108N: 86/359; N51I,C59R,S108N: 258/359), with an alternative less likely pathway of S108N/C59R/N51I (C59R,S108N: 6/359), corresponding to the thirteenth and first most likely trajectories to a triple mutation in our simulations, respectively. In South America, evolution up to the triple mutation N51I,S108N,I164L was observed in the isolate data and was inferred to follow the pathway S108N/N51I/I164L (S108N: 26/50, N51I,S108N: 12/50; N51I,S108N,I164L: 4/50), however this pathway does not occur in our simulations. In Melanesia, evolution up to the double mutation C59R,S108N was observed in the isolate data and was inferred using the frequency data to have followed the pathway S108N/C59R (S108N: 1/119; C59R,S108N: 23/119), which corresponds to the most likely pathway of all our evolutionary runs.

We performed an analysis of the significance of the regional frequency distributions (see Appendix 4 and *Appendix 4—figure 2*). All regions had mutations which were significantly over- or underrepresented compared to what would be expected from the worldwide distribution. The overrepresented mutations were always part of the inferred most likely evolutionary pathway for each region. This suggests any differences in the inferred pathways between the regions and the worldwide data are significant. For example, in South America (*Appendix 4—figure 2d*), the three mutations involved in the most likely inferred pathway (single mutation S108N, double mutation N51I,S108N, and triple mutation N51I,S108N,I164L) are all overrepresented. These mutations are involved in the most likely inferred pathway in that region (S108N/N51I/I164L), suggesting this region is indeed following a different evolutionary trajectory to what we would expect from the worldwide data.

Next, we considered the frequency of combinations of the four *Pv*DHFR mutations in different geographical regions and used these frequencies to infer evolutionary trajectories (*Figure 4*). As mentioned previously, the quadruple mutation is not observed in the isolate data and so we will infer trajectories up to triple mutant combinations of the four mutations where possible, and compare to the most likely pathways to triple mutations in our simulations and in the experimental data. Triple mutation S58R,S117N,I173L was the only triple mutant combination observed in the isolate data and was only found in South America. Analysing the frequency of the constituent mutations (S117N: 34/257; S58R,S117N: 74/257; S58R,S117N,I173L: 2/257), the most likely inferred pathway to this mutation is S117N/S58R/I173L. This was the fifth most likely triple mutation pathway in our simulations, however it was not an observed pathway in *Jiang et al., 2013*. There are two other possible pathways inferred from the frequency data to this triple mutation: S117N/I173L/S58R (S117N,I173L: 1/257) and S58R/S117N/I173L (S58R: 9/257). Pathway S117N/I173L/S58R was the third most likely pathway to a triple mutation in our simulations but was not observed in the simulations in *Jiang et al., 2013*, and S58R/S117N/I173L the most likely pathway to a triple mutation in our simulations and was observed as part of the second most likely pathway to the quadruple mutation at the highest pyrimethamine concentration considered in *Jiang et al., 2013*.

Evolution only up to double mutation S58R,S117N was found in Southeastern Asia, Eastern Africa, and Melanesia and S117N/S58R was the most likely fixation order in Eastern Africa, whilst S58R/S117N was the most likely fixation order in Southeastern Asia and Melanesia. Pathways S58R/S117N and S117N/S58R were the second and fourth most likely double mutation pathways in our simulations. In Southern Asia, evolution up to double mutations S58R,S117N and N50I,S117N was observed,

following pathways S117N/S58R and S117N/N50I, respectively (S58R: 3/37, S117N: 6/37, N50I,S117N: 6/37, S58R,S117N:22/37), with the pathway S117N/S58R appearing to be more prevalent. In Eastern Asia, evolution only up to single mutation S58R was observed and in Central America no steps in the evolutionary pathway including combinations of these four mutations were found.

We performed an analysis of the significance of the regional distributions, similar to described above for *Pf*DHFR, however the frequencies of the four *Pv*DHFR in many of the regions was too small to definitively draw conclusions (see Appendix 4 and *Appendix 4—figure 2* for more details). However, from this analysis it does appear that the distribution of mutations in South America is very similar to the worldwide distribution (*Appendix 4—figure 2d*) and this region is following the same inferred evolutionary pathway as the pathway inferred from the worldwide data (S117N/S58R/I173L). The distribution of mutations in Eastern Africa is similar to the worldwide distribution (*Appendix 4—figure 2a*), however this region is enriched for single mutation S117N and appears to have not evolved to the double mutant step in the most likely worldwide pathway (S58R,S117N) as frequently as would be expected, suggesting it is at an earlier stage of evolution compared to the worldwide distribution. Finally, double mutation N50I,S117N is overrepresented in Southern Asia (*Appendix 4—figure 2e*), suggesting an alternative evolutionary pathway may be occurring in this region.

## Discussion

We have presented a method for predicting the most likely evolutionary trajectories to multiple mutants by parameterising thermodynamic evolutionary model using Flex ddG predictions. The most likely pathways predicted by our model to the pyrimethamine-resistant quadruple *Pf*DHFR mutant correspond well to those predicted in *Lozovsky et al., 2009*, generated using experimentally determined IC50 values of *Pf*DHFR pyrimethamine binding. The two most likely pathways based on experimental IC50 values were found in the top two most likely pathways to the quadruple mutation based on our simulations using predictions of binding free energy. Whilst our simulations disagreed with the simulations in *Lozovsky et al., 2009*, in terms of which were the most frequently realised pathways out of the total number of runs, where a realised pathway does not necessarily have to reach the quadruple mutation, our model is able to capture the most likely order of fixation of mutations leading to a particular multiple mutant in general agreement with the simulations in *Lozovsky et al., 2009*.

We also simulated the most likely evolutionary trajectories to the *Pv*DHFR quadruple mutation N50I,S58R,S117N,I173L and compared our results to those predicted in *Jiang et al., 2013*. They considered the relative growth rates of the different alleles at different drug concentrations when simulating evolutionary trajectories, which incorporate both change in pyrimethamine binding affinity ($K_i$) and catalytic activity ($k_{cat}$). Our top two most likely pathways to the quadruple correspond to their top two most likely pathways for the highest pyrimethamine concentration they consider, albeit in reverse order. At high pyrimethamine concentrations, it is likely mutations that significantly reduce binding affinity will be selectively favoured even if there is a slight reduction in catalytic activity. Indeed, *Rodrigues et al., 2016*, observed a clear trade-off between catalytic activity and binding affinity for increased drug resistance. This may be why our predictions agree well their predictions for high pyrimethamine concentration, but not for low-to-middle pyrimethamine concentrations, because even though ligand concentration is included in our equation for protein fitness (*Equation 1*), our model cannot account for adaptive conflict between $K_i$ and $k_{cat}$. This highlights a limitation of our method as it only accounts for changes in binding affinity and does not account for changes in protein function.

Similarly, *Ogbunugafor et al., 2016*, demonstrated the importance of environmental variables on the evolution of drug resistance in *Pf*DHFR by simulating evolution considering empirical growth measures and IC50 values across drug concentrations. They found that the rank order of allele fitness is a function of drug concentration and the preferred pathways to the fittest allele depend upon the environment. Therefore, whilst the model presented here captures the most likely pathways to resistance for high drug concentration, it is unable to predict important evolutionary dynamics that arise as a function of the environment. Therefore, the pathways predicted here should be considered the most likely pathways under sustained use of high concentrations of pyrimethamine.

Several investigations have also studied reverse evolution in *Pf*DHFR (*Ogbunugafor, 2022*) and *Pv*DHFR (*Ogbunugafor and Hartl, 2016*) by simulating evolution from the resistant to susceptible phenotype for different pyrimethamine concentrations, including the drugless environment. These studies also used measurements of growth rates and IC50 at the different drug concentrations in their

simulations. They found S108N in *Pf*DHFR and S117N in *Pv*DHFR act as pivot point mutations and prevent reverse evolution to the susceptible phenotype. This has important consequences for AMR management strategies that aim to reduce resistance by the cessation of drug use. Unfortunately, our model cannot be used to study reverse evolution because the fitness function is unable to quantify fitness in a drug-free environment.

As previously mentioned, DHFR catalyses the reduction of substrate DHF via oxidation of co-factor NADPH. Therefore, in the case of the DHFR enzyme, a future iteration of the model could include the impact resistance mutations have on binding of these two ligands, as a proxy for changes to enzyme function. However, this would require a much more complex model of protein fitness and would be much more computationally expensive.

A further limitation of the evolutionary model is that it operates in the weak mutation regime, in which the mutation rate is so low that mutations appear and fix in isolation. However, this assumption breaks down when considering large microbial populations where clonal interference means that mutations can arise simultaneously and compete for fixation (*Gerrish and Lenski, 1998*). This can lead to a process known as 'greedy adaptation', whereby the mutation of largest beneficial effect out of all available mutations is fixed with certainty during an adaptive walk (*de Visser and Krug, 2014*). Interestingly, *Ogbunugafor and Eppstein, 2016*, simulated the competition between successive alleles along adaptive trajectories in the evolution of pyrimethamine resistance in both *Pf*DHFR and *Pv*DHFR, and found that greedy trajectories are not always the most likely to be followed, with less greedy trajectories sometimes able to reach higher fitness more quickly.

Clonal interference has been shown to emerge rapidly in laboratory cultivated *P. falciparum*, where the parasite cycles through only asexual stages, suggesting it may influence the dynamics of the emergence of resistance (*Jett et al., 2020*). The evolutionary model used here may therefore overestimate the fixation probability of mutations with milder beneficial effects and underestimate the fixation probability of mutations with larger beneficial effects (*de Visser and Krug, 2014*). Future iterations of this work could be improved by using a fixation probability that models clonal interference such as the work of *Gerrish and Lenski, 1998*, and *Campos et al., 2004*. However, such models are more difficult to implement as they can require species-specific derivations for certain functions.

Mutations occurring at a drug-binding site may also reduce the protein's thermodynamic stability (*Wang et al., 2002*) and therefore may not be selected for, even if they improve the resistance phenotype. However, our model does not take thermodynamic stability into account, and so may predict certain mutations to be beneficial to fitness even if they are actually deleterious because they decrease the protein's stability. This may lead to incorrect prediction of pathways. Our model may be improved by including selection for mutations that do not reduce thermodynamic stability relative to the wild-type enzyme, similar to the work by *Rotem et al., 2018*. However, it must also be noted that most proteins are marginally stable (*Vogl et al., 1997*; *Ruvinov et al., 1997*), a property which may have evolved either as an evolutionary spandrel (*Taverna and Goldstein, 2002*; *Goldstein, 2011*) (a characteristic that arises as a result of non-adaptive processes which is then used for adaptive purposes; *Gould and Lewontin, 1979*) or due to selection for increased flexibility to improve certain functionalities (*Závodszky et al., 1998*; *Tsou, 1998*). Therefore, the model would also need to account for the fact that a resistance mutation that increases protein stability relative to the wild-type stability may also result in a reduction in fitness.

In summary, the real fitness landscape of DHFR is much more complex than can capture with our model, which only accounts for changes in binding affinity. Therefore, our model may predict evolutionary pathways to be likely when they are in fact inaccessible if a step in the pathway is deleterious to other aspects of the fitness landscape.

Furthermore, our method uses Flex ddG to predict changes in drug binding free energy, which are then used to parameterise the fitness function of the evolutionary model. Therefore, the success of the method relies upon the accuracy of these predictions. As we established with the *Pf*DHFR mutations, Flex ddG performs well for the single mutations but is less accurate for the higher order mutations, meaning pathways may be predicted to be inaccessible by our model but which in reality are accessible. However, in general Flex ddG performed well and was able to capture the general trend of the data for the mutations considered here, and so the model showed good agreement with experimentally determined pathways. Our model could also be parameterised using molecular

dynamics, which may provide more accurate predictions of drug binding affinity, though with a much higher computational cost.

The main advantage of our model is its simplicity, requiring only predictions of change in drug binding affinity, to capture evolutionary pathways under the assumption of sustained high drug concentration.

Despite the limitations of our computational method to predict evolutionary trajectories by only considering the impact on drug binding, it is able to accurately predict the most likely order of fixation of mutations in a trajectory in general agreement with trajectories determined using experimental values such as IC50 or more complex fitness landscapes informed by multiple parameters including drug concentration and growth rates. This suggests evolution in such landscapes is more predictable than might be expected, since trajectories can be predicted considering only the impact on binding affinity.

We also inferred evolutionary pathways from the total frequency in worldwide clinical isolate data as well as from different geographical regions. This analysis suggests evolutionary pathways may be inferred from the frequency of mutations found in isolate data, however it requires a large number of isolates to properly sample the mutations in the population. Furthermore, this method can only be used to predict trajectories once resistance has emerged in a population, whereas the computational method presented here can predict evolutionary trajectories before introduction of a new drug.

This analysis also suggested that different regions often follow different evolutionary trajectories and that the most likely evolutionary trajectories predicted by our model, and experimental trajectories, are not always the most prevalent. Geographical differences in the distribution of resistant alleles may be the result of drug regimens and gene flow in parasite populations. Combination drug SP was first used in 1967 to treat *P. falciparum* in Southeastern Asia, and resistance was first noted that same year on the Thai-Cambodia and Thai-Myanmar borders (*Björkman and Phillips-Howard, 1990*). In Africa, SP was first used in the 1980s, with resistance occurring later that decade. However, analysis of *Pf*DHFR genotypes and microsatellite haplotypes surrounding the DHFR gene in Southeastern Asia and Africa suggest a single resistant lineage that appeared in Southeastern Asia accumulated multiple mutations, including the triple N51I,C59R,S108N (*Roper et al., 2004*; *Mita et al., 2007*), migrated to Africa and spread throughout the continent (*Maïga et al., 2007*; *McCollum et al., 2008*; *McCollum et al., 2007*). Variation in the frequency of *Pf*DHFR mutants across Africa occurs because of differences in the timing of chloroquine withdrawal and introduction of SP, as well as continued use of SP for intermittent preventive treatment (IPTp) in pregnant women residing in areas of moderate-to-high malaria transmission intensity (*Turkiewicz et al., 2020*; *Ravenhall et al., 2016*).

Pyrimethamine resistance increased in West Papua in the early 1960s following the introduction of mass drug administration (*Verdrager, 1986*). Microsatellite haplotype analysis suggests C59R,S108N in Melanesia has two lineages, one of which originated in Southeastern Asia whilst the other evolved indigenously (*Roper et al., 2004*).

Pyrimethamine resistance in South America looks surprisingly different from the distributions in Africa and Southeastern Asia. SP was introduced in South America and low-level resistance was first noted in Colombia in 1981 (*Espinal et al., 1985*). Microsatellite haplotype analysis suggests pyrimethamine resistance evolved indigenously in South America, with at least two distinct lineages detected. A triple mutant lineage (C50I,N51I,S108N) was identified in Venezuela that possibly evolved from double mutant N51I,S108N (*McCollum et al., 2007*). A second triple mutant lineage (N51I,S108N,I164L) was identified in Peru and Bolivia which also possibly evolved from a distinct double mutant (N51I,S108N) lineage (*Zhou et al., 2008*).

In general, the *Pv*DHFR gene is much more polymorphic than *Pf*DHFR gene, with over 20 alleles observed in a limited geographical sampling (*Hawkins et al., 2007*), whereas fewer *Pf*DHFR alleles have been observed despite much more extensive surveillance with non-synonymous changes and insertions/deletions occurring rarely (*Gregson and Plowe, 2005*). It also appears that the origin of *Pv*DHFR pyrimethamine resistance mutation is much more diverse than *Pf*DHFR. *Hawkins et al., 2008*, investigated isolates from Colombia, India, Indonesia, Papua New Guinea, Sri Lanka, Thailand, and Vanuatu and found multiple origins of the double *Pv*DHFR mutant 58R,117N in Thailand, Indonesia, and Papua New Guinea/Vanuatu. *Shaukat et al., 2021* assessed the resistance mutations in Punjab, Pakistan, and found multiple origins of single mutation S117N and a common origin of double mutant 58R,S117N and triple mutant 58R,117N,I173L. This is in contrast to the evolutionary origin of

pyrimethamine resistance in *Pf*DHFR, where mutations in Africa shared a common origin with a resistance lineage from Asia.

This highlights the need to distinguish between geographical regions and account for existing resistance alleles within that region and trace their lineages when attempting to predict the next step in evolutionary trajectories to highly resistant multiple mutants. Given the current dominant resistance allele from a specific region, our method could be used to predict the most likely next steps from a subset of likely mutations.

We have presented a computational method for predicting the most likely evolutionary trajectories that has demonstrated good agreement with trajectories predicted experimentally and has the advantage of being much quicker and more cost-effective than laboratory-based methods. This method can be applied to any system in which a drug binds to a target molecule, provided a structure of the complex exists or can be produced via structural modelling. Given the threat AMR poses, methods to accurately and efficiently predict future trajectories are vital and can inform treatment strategies and aid drug development.

# Materials and methods
## Homology modelling
Homology modelling was carried out in Modeller (*Webb and Sali, 2016*) to produce complete structures of the target proteins bound to their drug molecules. Several crystal structures of *Pf*DHFR exist in the Protein Data Bank (PDB). The entry 3QGT provides the crystal structure of wild-type *Pf*DHFR complexed with NADPH, dUMP, and pyrimethamine, however residues in the ranges 86–95 and 232–282 are missing from the structural model. Homology modelling was used to complete the structure using a second wild-type *Pf*DHFR structure PDB entry 1J3I along with a wild-type *Pv*DHFR structure PDB entry 2BLB.

To produce a complete structural model of *Pv*DHFR, PDB entry 2BLB was used as a template, which provides the X-ray crystal structure of wild-type *P. vivax* DHFR in complex with pyrimethamine. This structure was only missing a loop section between residues 87–105 and so Modeller was used to build this missing loop.

## Flex ddG binding free energy predictions
The Rosetta Flex ddG protocol was used to estimate the change in binding free energy upon mutation, $\Delta\Delta G = \Delta G_{mut} - \Delta G_{WT}$, for each step in all possible mutational trajectories for a set of step-wise resistance mutations (see Supplementary data Flex_ddG folder for examples of a Rosetta script, resfile, and command line. The protein-ligand structure files and ligand parameter files can be found in the folders named for the specific targets). To predict the change in binding free energy for a single or multiple mutation, we used the structure of the target protein with the drug molecule bound as input to Flex ddG and ran the protocol for 250 times per mutation to produce a distribution of predictions of the change in the free energy of binding. We then found the mean of the distribution to produce a single estimate of the change in the binding free energy for the mutation, denoted $\Delta\Delta G_X^*$ for mutation $X$.

To predict the stepwise evolutionary trajectories, we must consider the interactions between the mutations in the pathway. The interaction energy (or epistasis) in the binding free energy between two mutations $X$ and $Y$ can be written $\epsilon_{X,Y} = \Delta\Delta G_{X,Y} - (\Delta\Delta G_X + \Delta\Delta G_Y)$. This quantifies by how much the change in binding free energy of the double mutant $X,Y$ deviates from additivity of the single mutants, where each are calculated with respect to the wild-type. Therefore, the change in binding free energy when mutation $Y$ occurs in the background of mutation $X$ can be written $\Delta\Delta G_{X/Y} = \Delta\Delta G_{X,Y} - \Delta\Delta G_X$, where $\Delta\Delta G_{X/Y} = \Delta\Delta G_Y + \epsilon_{X,Y}$.

For a third mutation, $Z$, occurring in the background of double mutation $X,Y$, the interaction energy between $Z$ and $X,Y$ is $\epsilon_{XY,Z} = \Delta\Delta G_{X,Y,Z} - (\Delta\Delta G_{X,Y} + \Delta\Delta G_Z)$. The quantity $\epsilon_{XY,Z}$ is not the same as the third-order epistasis between mutations $X$, $Y$, and $Z$, or the interaction energy $\epsilon_{XYZ} = \Delta\Delta G_{X,Y,Z} - (\Delta\Delta G_X + \Delta\Delta G_Y + \Delta\Delta G_Z)$ as it does not account for the interaction between $X$ and $Y$, rather it only quantifies the interaction between $Z$ and the two mutations $X$ and $Y$. Therefore, the change in binding free energy when mutation $Z$ occurs in the background of double mutant $X,Y$ can be calculated as $\Delta\Delta G_{X,Y/Z} = \Delta\Delta G_{X,Y,Z} - \Delta\Delta G_{X,Y}$, where $\Delta\Delta G_{X,Y/Z} = \Delta\Delta G_Z + \epsilon_{XY,Z}$.

To estimate the change in binding free energy when mutation $Y$ occurs in the background of mutation $X$, $\Delta\Delta G_{X/Y}$ for stepwise pathway $X/Y$, we subtracted the predictions $\Delta\Delta G_X^i$ for the first mutation $X$, from the predictions for the double mutation $X, Y$, $\Delta\Delta G_{X,Y}^i$, to create a set of 250 'predictions' for the change in binding free energy when $Y$ occurs in the background of $X$, $\Delta\Delta G_{X/Y}^i$, i.e., $\Delta\Delta G_{X/Y}^i = \Delta\Delta G_{X,Y}^i - \Delta\Delta G_X^i$ for $i = \{1, ..., 150\}$. To estimate the change in binding free energy when mutation $Z$ occurs in the background of mutations $X$ and $Y$, we calculated $\Delta\Delta G_{X,Y/Z}^i = \Delta\Delta G_{X,Y,Z}^i - \Delta\Delta G_{X,Y}^i$. We applied a similar method for the quadruple mutations, so that we had a set of 'predictions' for each step in the possible evolutionary trajectories.

## Simulating evolutionary trajectories

The Rosetta energy function is a mix of a combination of physic-based and statistics-based potentials and so raw predictions using this function don't match up with physical energy units (e.g. kcal/mol or kJ/mol). However, the authors of Flex ddG applied a generalised additive model-like approach to the Rosetta energy function to reweight its terms and to fit experimentally known values (in kcal/mol). The resulting nonlinear reweighting model reduced the absolute error between the predictions and experimental values and so improved the agreement with experimentally determined interface $\Delta\Delta G$ values. They found that by doing this the Flex ddG predictions of binding free energy changes were in a similar range as experimental binding free energy changes and observed improved correlation and classification of mutations as stabilising or destabilising (*Barlow et al., 2018*). Therefore, we assume Flex ddG can provide approximate predictions of binding free energy changes comparable to experimental changes in kcal/mol and can therefore be used to parameterise a thermodynamic model.

To predict the most likely evolutionary trajectories to reach a quadruple mutant, we used a model based in thermodynamics and statistical mechanics where the fitness of a protein is determined by the probability it would not be bound to a ligand, $P_{unbound}$. We consider a two-state system in which the protein can either be bound or unbound and do not explicitly account for if the protein is folded or unfolded in either the bound or unbound state. For ligand concentration $[L]$ it can be shown that the probability a protein is unbound is

$$P_{unbound} = \frac{1}{\frac{[L]}{K_d} + 1} \tag{1}$$

where $K_d$ is the protein-ligand dissociation constant and can be calculated as $c_0 e^{\Delta G/kT}$, where $c_0$ is a reference ligand concentration (set here arbitrarily to 1 M), $\Delta G$ is the protein-ligand binding free energy, $k$ is the Boltzmann constant, and $T$ is the temperature in Kelvin. This equation is derived under the assumption that the concentration of unbound ligand $L_{free} \approx L_0$, where $L_0$ is the total concentration. This captures the ideal scenario in which the concentration of the antimalarial drug is very high. Whilst resistance is trade-off between many biophysical factors, our aim is to predict evolutionary trajectories without the need for experimental measurements and ligand binding prediction tools are readily available and easy to use, whereas it is more difficult to predict IC50 or growth rates. We therefore chose to use a simplified model of the system to approximate its behaviour, capture its general properties, and make the model more tractable.

Starting from the wild-type protein, with binding free energy $\Delta G_{WT}$ and fitness $P_{unbound}^{WT}$, we extract one sample $i$ from the 250 values of the predicted binding affinity changes for the single mutations to determine the binding free energy after mutation $X$, $\Delta G_X^i = \Delta G_{WT} + \Delta\Delta G_X^i$, and calculate the fitness of each single mutant protein $P_{unbound}^{X(i)}$. We can calculate the probability the mutation will fix in the population using the Kimura fixation probability for a haploid organism

$$p_{fix} = \frac{1 - e^{-2s}}{1 - e^{2sN_e}} \tag{2}$$

where $N_e$ is the effective population size (set to $10^6$ as previous models in *Eccleston et al., 2021*; *Pollock et al., 2012*) and $s$ is the selection coefficient $s = (P_{unbound}^{X,i} - P_{unbound}^{WT})/P_{unbound}^{WT}$. We also took into account the mutational bias of *P. falciparum* using the nucleotide mutation matrix calculated in *Lozovsky et al., 2009*. The probabilities of fixation for each mutation were normalised by the sum of the probabilities of fixation for all possible mutations at that step in the trajectory. A mutation is then chosen with a probability proportional to this normalised probability of fixation.

Once a single mutation is chosen, the binding free energy is set to $\Delta G_X^i$ of the chosen mutation, and a value is sampled from the distribution of each of the possible next steps, $X/Y$ in the trajectory, i.e., $\Delta\Delta G_{X/Y}^i$. This continues until the end of the trajectory is reached. If the fixation probabilities of all mutations sampled at a step are effectively zero, no mutation is chosen at that step and the algorithm begins again by choosing a single mutation. Therefore, not all of the runs produce a complete trajectory and some will terminate before reaching the quadruple mutation. The algorithm was written in *R*.

We calculate the probabilities, $P_i$, of each realised pathway (even those which don't reach the quadruple mutation) by dividing the total number of times that specific pathway occurs by the number of runs. We calculate the probability of a particular step by dividing the number of times that step occurs in all realised pathways by the total number of runs.

### SNP data

*P. falciparum* and *P. vivax* data was obtained from publicly available raw sequence data from European Nucleotide Archive. These data include Illumina raw sequences from the MalariaGEN Community Project for *P. falciparum* (***Ahouidi et al., 2021***) and *P. vivax* (***Adam et al., 2022***). *P. vivax* data additionally includes the Public Health England Malaria Reference Laboratory isolates from returning travelers to UK from regions where malaria is endemic (study accession number ERP128476) (***Benavente et al., 2021***). Data was filtered and processed to SNP data with the methodology described in the recent publications ***Turkiewicz et al., 2020***, and ***Benavente et al., 2021***, respectively. In this study, we analysed genotype data for 6762 high-quality isolates from 32 countries across regions of Africa, South Eastern Asia, Oceania, and South America to identify the genetic diversity in the *Pf*DHFR gene. A similar analysis was carried out on 847 *P. vivax* isolates spanning 25 countries across Eastern Africa, Southern Asia, Southeastern Asia, Eastern Asia, and South America to identify genetic diversity in *Pv*DHFR gene. SNPs occurring in non-unique, low-quality, or low coverage regions were discarded, and those with a missense effect in the candidate genes were analysed. Functional annotation was done with *SnpEff* (version 5.0) (***Cingolani et al., 2012***) with the following options: *-no-downstream -no-upstream*.

## Acknowledgements

RCE and NF are funded by the Medical Research Council UK (Grant no. MR/T000171/1). TGC is funded by the Medical Research Council UK (Grant no. MR/M01360X/1, MR/N010469/1, MR/R025576/1, MR/R020973/1, and MR/T000171/1) and BBSRC (Grant no. BB/R013063/1). SC is funded by Medical Research Council UK grants (MR/M01360X/1, MR/R025576/1, and MR/R020973/1) and Bloomsbury SET. EM is funded by a Newton Institutional Links Grant British Council, no. 261868591. The funders had no role in study design, data collection, and analysis, decision to publish, or preparation of the manuscript. RCE would like to thank Tanushree Tunstall (London School of Hygiene and Tropical Medicine) for the helpful discussions. RCE would also like to thank Professor Richard Goldstein (University College London) for introducing her to the evolutionary algorithm used in this paper.

## Additional information

### Funding

| Funder | Grant reference number | Author |
| --- | --- | --- |
| Medical Research Council | MR/T000171/1 | Ruth Charlotte Eccleston<br>Nicholas Furnham<br>Taane G Clark |
| Medical Research Council | MR/M01360X/1 | Taane G Clark<br>Susana Campino |
| Medical Research Council | MR/N010469/1 | Taane G Clark |
| Medical Research Council | MR/R025576/1 | Taane G Clark<br>Susana Campino |

| Funder | Grant reference number | Author |
| --- | --- | --- |
| Medical Research Council | MR/R020973/1 | Taane G Clark<br>Susana Campino |
| Biotechnology and Biological Sciences Research Council | BB/R013063/1 | Taane G Clark |
| British Council | 261868591 | Emilia Manko |

The funders had no role in study design, data collection and interpretation, or the decision to submit the work for publication.

## Author contributions
Ruth Charlotte Eccleston, Conceptualization, Formal analysis, Investigation, Methodology, Writing - original draft, Writing - review and editing; Emilia Manko, Conceptualization, Data curation, Writing - review and editing; Susana Campino, Conceptualization, Data curation; Taane G Clark, Conceptualization; Nicholas Furnham, Conceptualization, Writing - review and editing

## Author ORCIDs
Ruth Charlotte Eccleston ⬡ http://orcid.org/0000-0002-1905-7942
Taane G Clark ⬡ http://orcid.org/0000-0001-8985-9265
Nicholas Furnham ⬡ http://orcid.org/0000-0002-7532-1269

## Decision letter and Author response
Decision letter https://doi.org/10.7554/eLife.84756.sa1
Author response https://doi.org/10.7554/eLife.84756.sa2

# Additional files

## Supplementary files
• MDAR checklist

## Data availability
Code, Rosetta Flex ddG predictions, structural models and isolate mutation frequency data deposited in Zenodo at https://doi.org/10.5281/zenodo.7082168.

The following dataset was generated:

| Author(s) | Year | Dataset title | Dataset URL | Database and Identifier |
| --- | --- | --- | --- | --- |
| Charlotte ER, Emilia M, Susana C, Taane GC, Nicholas F | 2023 | A computational method for predicting the most likely evolutionary trajectories in the stepwise accumulation of resistance mutations | https://doi.org/10.5281/zenodo.7082168 | Zenodo, 10.5281/zenodo.7082168 |

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

## Appendix 1

### Comparison of Flex ddG predictions for 35–250 runs

For the set of four *Pf*DHFR mutations (N51I, C59R, S108N, and I164L) and combinations thereof studied in *Lozovsky et al., 2009*, we ran 35 runs per mutation (as suggested by the Flex ddG authors; *Barlow et al., 2018*) and found the average for each distribution and used these predictions to calculate the sum and the interaction energies for the multiple mutants. We compared the predictions of the binding free energy change, the sum of the independent changes for multiple mutants, and the interaction energy to the experimental data from *Lozovsky et al., 2009*, and observed Pearson correlations of 0.536, 0.580, and 0.900, respectively (*Appendix 1—table 1*). We also determined the number of correctly classified predictions. For the binding free energy predictions, 5/9 predictions were correctly classified as stabilising or destabilising, 4/5 of the sum of the independent impacts were correctly classified as stabilising or destabilising, and 2/5 interaction energy predictions were correctly classified as either positive or negative. Therefore, whilst the predictions for 35 runs achieved a good correlation with the data, the predictions of the interaction energy (and so the epistasis) using this data were correctly classified for less than half of the dataset.

Examining the distributions of the predicted change in binding free energy for 35 runs for each of the mutations considered in *Lozovsky et al., 2009* (*Appendix 1—figure 1*), we can see that the distributions are not well characterised. We therefore decided to carry out a larger number of runs per prediction and determine the number of runs required for the rank order of the mutations to converge. We found the rank order of the average of the distributions sufficiently converged by 250 runs (*Appendix 1—figure 3*), as demonstrated in *Appendix 1—figure 4* where the gradient of the average for each mutation is close to zero. Whilst more runs may have achieved better convergence, because of the time it takes to run Flex ddG it is important to achieve a balance between efficiency and accuracy. Whilst these new distributions are still not Gaussian, the distributions are better explored (*Appendix 1—figure 2*). We compared the predictions for 250 runs and the data from *Lozovsky et al., 2009* (*Table 1*) and observed a correlation of 0.611 for the binding free energy data, 0.660 for the sum of the independent predictions for multiple mutants, and 0.756 for the interaction energy. We found 8/9 binding free energy predictions were correctly classified, 4/5 of the sum of the independent predictions were correctly classified, and 4/5 of the interaction energies were correctly classified. We therefore conclude that the predictions for *n*=250 runs present a better agreement with the data presented in *Lozovsky et al., 2009*, in terms of compromising between correlation and correct classification, both of which are important here.

**Appendix 1—table 1.** Correlation between Flex ddG predictions for 35 runs and experimental data (see Table 4 of *Sirawaraporn et al., 1997*) for *P. falciparum* dihydrofolate reductase (*Pf*DHFR) pyrimethamine resistance mutations.

| Mutation | $\Delta\Delta G_{exp}$*(kcal/mol) | Exp. sum† | Exp I.E. ‡ | $\Delta\Delta G_{FlexddG}$ §(kcal/mol) | Sum¶ | I.E.§** |
|---|---|---|---|---|---|---|
| N51I | –0.783 | | | –0.156 | | |
| C59R | –0.184 | | | 0.059 | | |
| S108N | 1.297 | | | 0.521 | | |
| I164L | –0.351 | | | –0.661 | | |
| N51I,S108N | 1.89 | 0.514 | 1.376 | –0.132 | 0.365 | –0.497 |
| C59R,S108N | 2.29 | 1.113 | 1.177 | 0.399 | 0.580 | –0.183 |
| N51I,C59R,S108N | 2.595 | 0.33 | 2.265 | 0.196 | 0.425 | –0.228 |
| C59R,S108N,I164L | 3.283 | 0.762 | 2.521 | –0.004 | –0.081 | 0.077 |
| N51I,C59R,S108N,I164L | 3.761 | –0.021 | 3.782 | 0.306 | –0.237 | 0.542 |
| Pearson correlation | | | | 0.536 | 0.580 | 0.900 |
| Correctly classified | | | | 5/9 | 4/5 | 2/5 |

*Appendix 1—table 1 Continued on next page*

*Appendix 1—table 1 Continued*

| Mutation | $\Delta\Delta G_{exp}$*(kcal/mol) | Exp. sum† | Exp I.E. ‡ | $\Delta\Delta G_{FlexddG}$ §(kcal/mol) | Sum¶ | I.E.§** |
|---|---|---|---|---|---|---|

*Experimentally measured *Pf*DHFR pyrimethamine binding free energy change data from ***Sirawaraporn et al., 1997***.

†Sum of experimental values of binding free energy change for independent mutations.

‡Interaction energy calculated as the difference between experimentally measured values of binding free energy change of multiple mutant compared to the sum of the independent mutations involved.

§Change in *Pf*DHFR pyrimethamine binding free energy predicted by Flex ddG calculated as the average of the distribution of runs. Free energy predictions from Rosetta are in Rosetta Energy Units, however the authors of Flex ddG applied a generalised additive model to reweight the predictions and make the output more comparable to units of kcal/mol (***Barlow et al., 2018***).

¶Sum of Flex ddG predictions for independent mutations

**Interaction energy calculated as the difference between Flex ddG predicted binding free energy change of multiple mutant compared to the sum of the independent mutations.

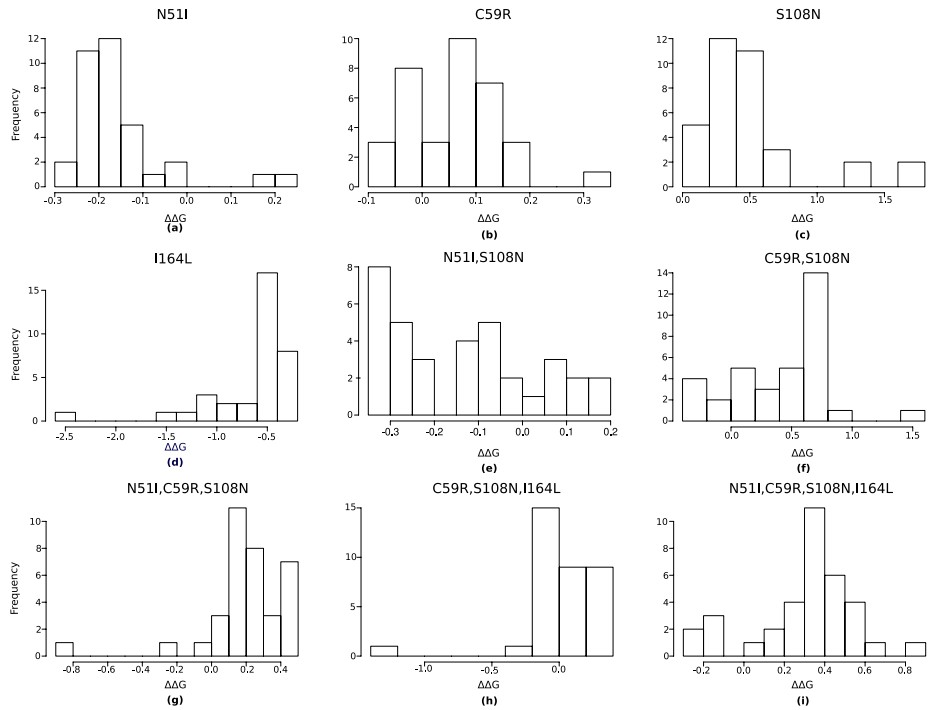

**Appendix 1—figure 1.** The distribution of pyrimethamine-*P. falciparum* dihydrofolate reductase (*Pf*DHFR) binding free energy changes predicted by Flex ddG for 35 runs for subset of mutations considered in ***Sirawaraporn et al., 1997***, namely (**a**) N51I, (**b**) C59R, (**c**) S108N, (**d**) I164L, (**e**) N51I,S108N, (**f**) C59R,S108N, (**g**) N51I,C59R,S108N, (**h**) C59R,S108N,I164L, and (i) N51I,C59R,S108N,I164L.

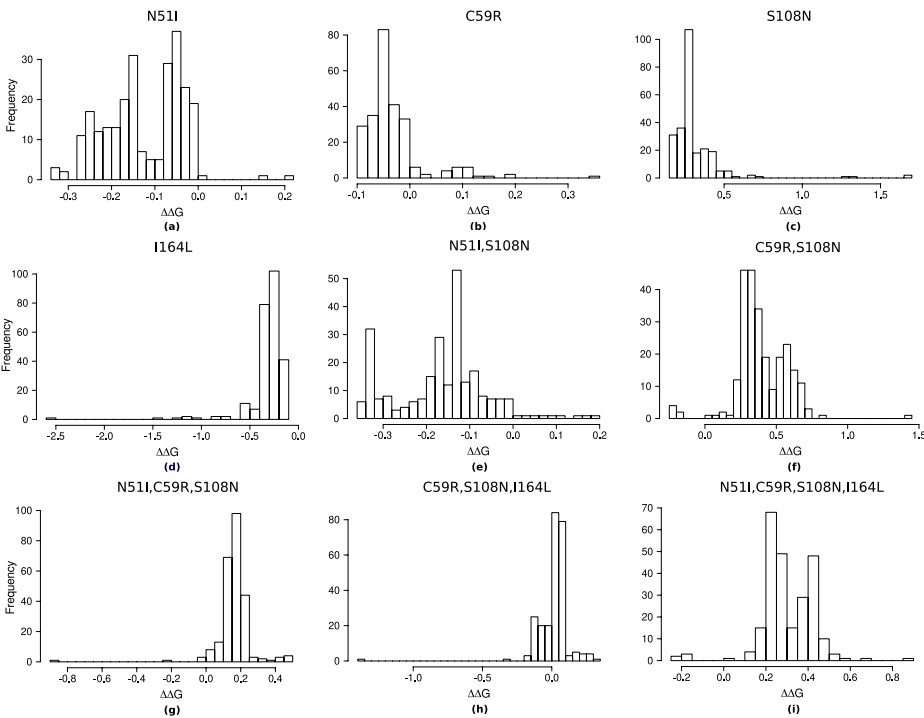

**Appendix 1—figure 2.** The distribution of pyrimethamine-*P. falciparum* dihydrofolate reductase (*Pf*DHFR) binding free energy changes predicted by Flex ddG for 250 runs for subset of mutations considered in ***Sirawaraporn et al., 1997***, namely (**a**) N51I, (**b**) C59R, (**c**) S108N, (**d**) I164L, (**e**) N51I,S108N, (**f**) C59R,S108N, (**g**) N51I,C59R,S108N, (**h**) C59R,S108N,I164L, and (**i**) N51I,C59R,S108N,I164L.

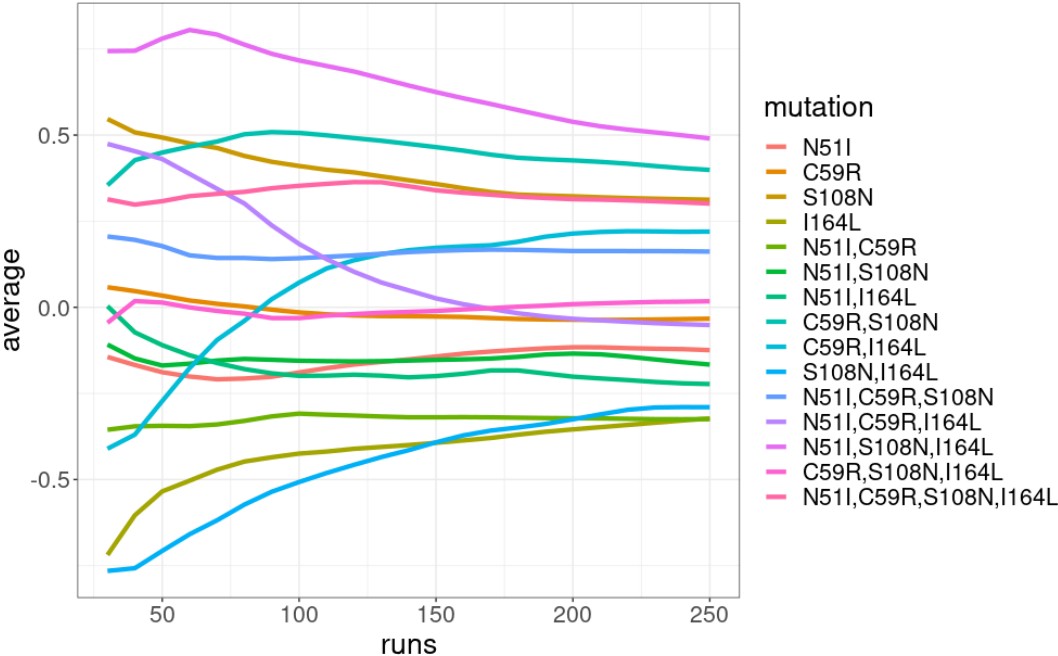

**Appendix 1—figure 3.** The running average of the Flex ddG prediction distributions for *n*=30,…,250 runs in intervals of 10 for the *P. falciparum* dihydrofolate reductase (*Pf*DHFR) mutation combinations.

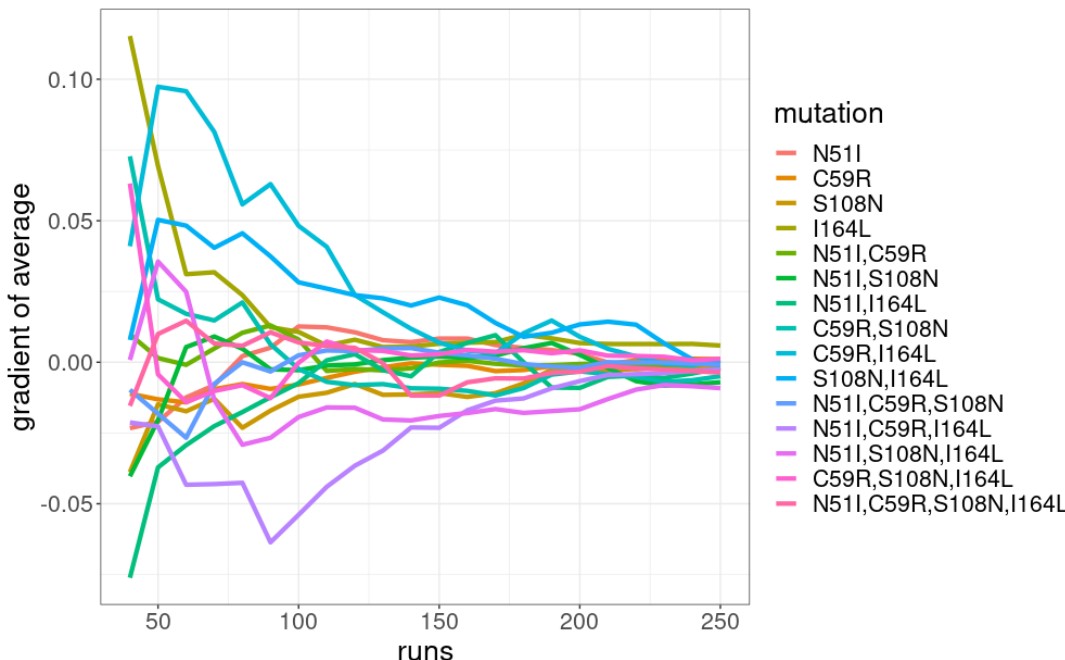

**Appendix 1—figure 4.** The gradient of the running average of the Flex ddG prediction distributions for $n=30,\ldots,250$ runs in intervals of 10 for the *P. falciparum* dihydrofolate reductase (*Pf*DHFR) mutation combinations.

## Appendix 2

### *Pf*DHFR fitness hypercube

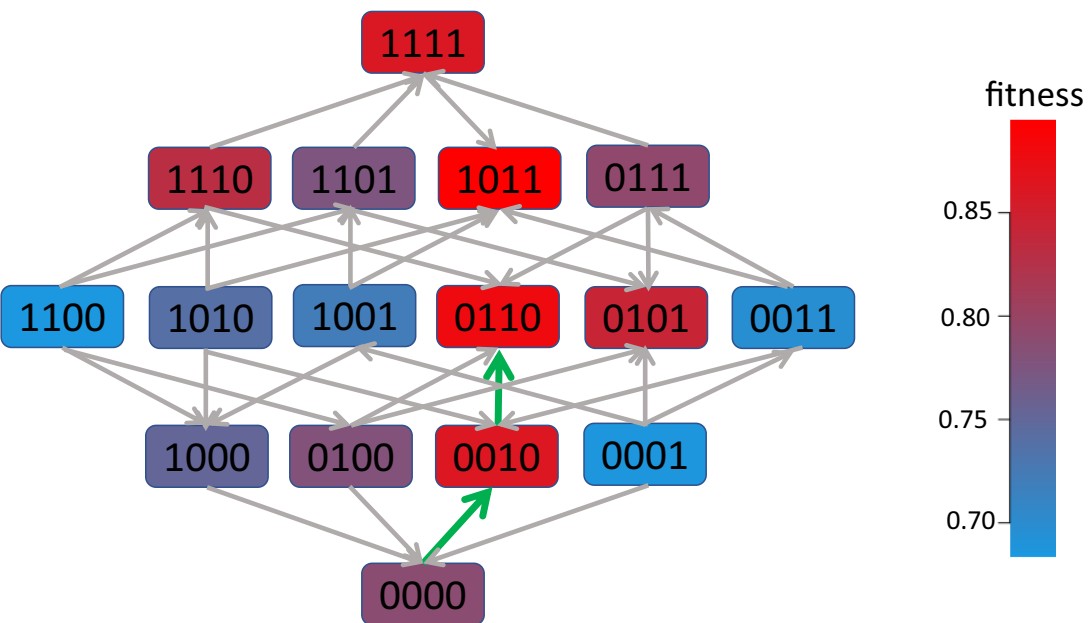

**Appendix 2—figure 1.** Fitness hypercube for combinatorically complete set of $2^4$ *P. falciparum* dihydrofolate reductase (*Pf*DHFR) genotypes. Paths of increasing fitness are shown inn green. Grey lines depict increasing fitness between genotypes separated by a single mutation. The fitness of each genotype is predicted using the average of each Flex ddG distribution to parameterise *Equation 1*.

## Appendix 3

## Assessing the significance of the frequency distribution of mutations in the worldwide isolate data

We performed a Chi-squared test on the worldwide distribution of the four *Pf*DHFR mutations considered here, to determine if the mutations involved in the most likely inferred pathway were overrepresented compared to what we would expect under a null hypothesis in which there is no preferred pathway. Under this null hypothesis, if there was no preferred pathway, the single mutations would be observed at the same frequency, the double mutations would be observed at the same frequency, and the triple mutations would be observed at the same frequency. Therefore, we carried out three separate Chi-squared tests on the distributions of the single, double, and triple mutations. (It would be difficult to carry out a Chi-squared test on the combined distribution of these mutations because we would not expect the single, double, and triple mutations to have the same frequency and it would be difficult to determine what their appropriate relative frequencies would be.) The Chi-squared tests determined all three distributions were significantly different from the null hypothesis ($p_{singles} < 0.01$, $p_{doubles} = 0$, $p_{triples} = 0$). Analysing the residuals of each distribution (*Appendix 3—figure 1*), S108N was found to be overrepresented in the distribution of single mutations, C59R,S108N was overrepresented in the distribution of double mutations, and N51I,C59R,S108N was overrepresented in the triple mutation distribution. These mutations are the single, double, and triple mutations involved in the most likely inferred pathway for the worldwide data, supporting our assertion that this is the most likely stepwise trajectory to the quadruple mutation.

We performed a Chi-squared test for the distribution of the *Pv*DHFR mutations in a similar way to *Pf*DHFR, however the frequency of triple mutants in the *Pv*DHFR worldwide dataset was not large enough to accurately carry out the test on the distribution of triple mutations. Therefore, we only carried out the test for the distribution of single and double mutations (*Appendix 3—figure 1*), with the null hypothesis that if there were no preferred order of fixation, the frequency of the single mutations would be equal and the frequency of the double mutations would be equal. The Chi-squared test revealed the distribution of both the single and double mutants is significantly different from what we would expect from the null hypothesis ($p_{singles} < 0.01$ and $p_{doubles} < 0.01$ for the single and double distributions, respectively). Analysing the residuals of each distribution, S117N was found to be overrepresented among the single mutations and S58R,S117N was found to be overrepresented among the double mutations. This provides support for the idea that epistasis determines the order of fixation and suggests that the most likely pathway to the only triple mutation observed (S58R,S117N,I173L) occurs via pathway S117N/S58R/I173L. This corresponds to the fifth most likely pathway to a triple mutation when considering all pathways observed in our simulations, however this pathway is not observed in any of the most frequent pathways at any of the four concentrations studied in *Jiang et al., 2013*.

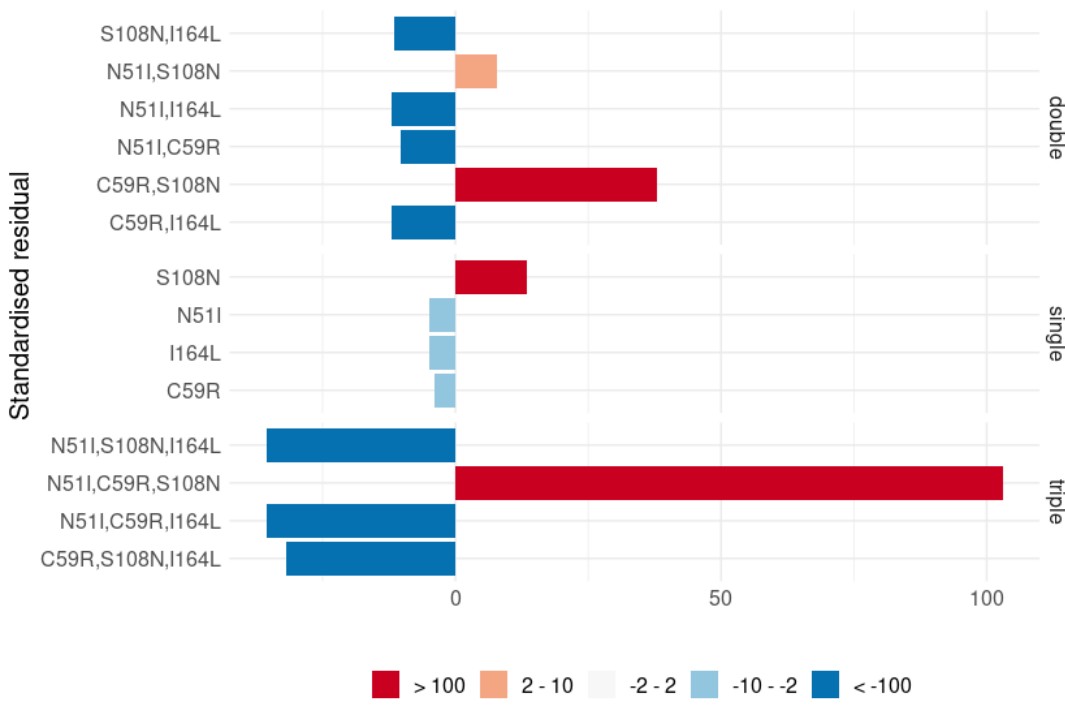

**Appendix 3—figure 1.** The standardised residuals of the individual *P. falciparum* dihydrofolate reductase (*Pf*DHFR) mutations from the Chi-squared tests applied to single, double, and triple mutants.

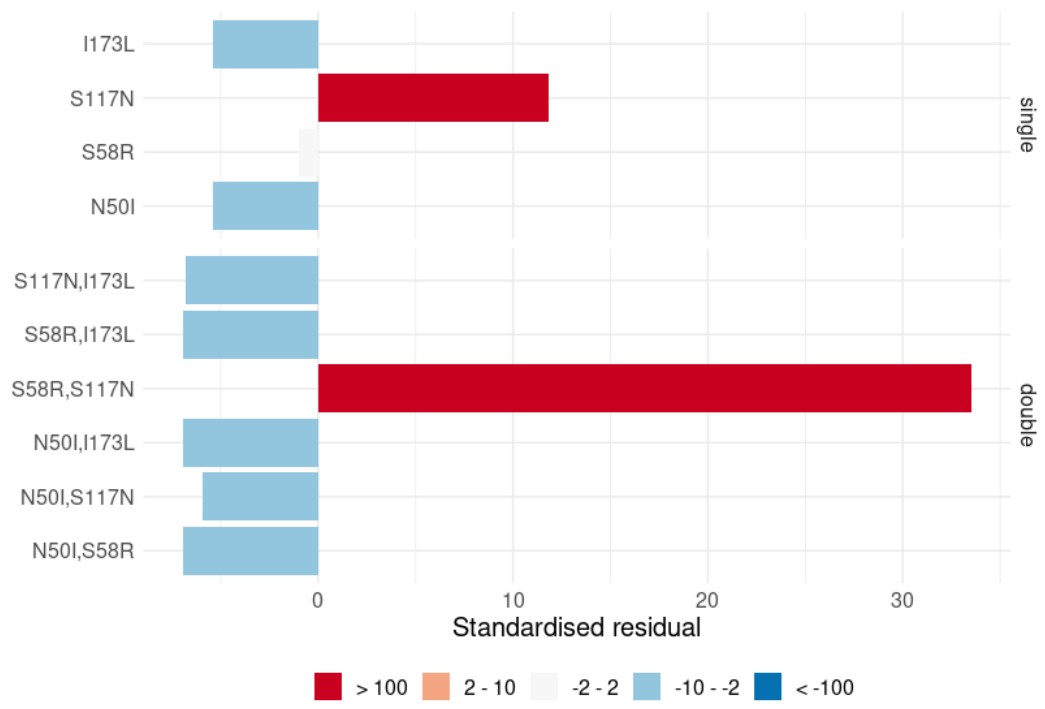

**Appendix 3—figure 2.** The standardised residuals of the individual *P. vivax* dihydrofolate reductase (*Pv*DHFR) mutations from the Chi-squared tests applied to single and double mutants.

# Appendix 4

## Assessing the significance of the frequency distribution of mutations per region

The frequency of the mutations in the separate regions is often too small to reliably carry out a Chi-squared test to determine if the regional distributions are significantly different from the worldwide distribution. Therefore, for each region, a sample of size $N$ (where $N$ is the size of the dataset from that region) was drawn with replacement from the worldwide distribution, and this was repeated 50,000 per region to create a dataset of 50,000 bootstrap samples per region. Comparing the frequency of mutations in the samples to the regional data highlights those mutations whose frequency differs significantly from what would be expected from the worldwide distribution.

### *Pf*DHFR

The distribution of mutations in Western Africa is similar to the worldwide distribution with the exception of mutations N51I,C59R,S108N and N51I,C59R,S108N,I164L which are overrepresented and underrepresented in the region, respectively (*Appendix 4—figure 1g*). This suggests the region is following the same pathway as the worldwide distribution but that the evolution is at an earlier stage. Conversely, in Southeastern Asia (*Appendix 4—figure 1f*), N51I,C59R,S108N was underrepresented in the region, whilst N51I,C59R,S108N,I164L was overrepresented, suggesting the region is following the same pathway as the worldwide distribution but evolution to the quadruple mutation had occurred more often than expected. Analysis of the distribution of mutations in Southern Asia (*Appendix 4—figure 1e*) suggests double mutation C59R,S108N is overrepresented in this region and the evolution in this region is more concentrated around the double mutant step in the pathway than would be expected. Triple mutations N51I,C59R,S108N and C59R,S108N,I164L are under- and overrepresented in this region, respectively, suggesting the alternative pathway S108N/C59R/I164L/N51I is more prevalent in this region than expected.

The distribution of mutations in Eastern Africa (*Appendix 4—figure 1b*) is similar to the worldwide distribution with the exception of double mutations N51I,S108N and C59R,S108N which are slightly over- and underrepresented in the region, respectively. Furthermore, triple mutation N51I,C59R,S108N and the quadruple mutation are over- and underrepresented, respectively. This suggests a true preference for the double mutant step S108N/N51I over S108N/C59R in the pathway in this region compared to the worldwide distribution. Furthermore, similar to Western Africa, the overrepresentation of the triple mutant step in the most likely inferred pathway and the underrepresentation of the quadruple mutation suggests this region is at an earlier stage in evolution compared to what would be expected from the worldwide distribution.

Similar to Eastern Africa, the distribution of mutations in Middle Africa (*Appendix 4—figure 1a*) showed significant differences in the frequency of double mutations N51I,S108N and C59R,S108N which were over- and underrepresented, respectively. Triple mutation N51I,C59R,S108N was overrepresented in this region and the quadruple mutation was underrepresented. This suggests that like Eastern Africa, the evolutionary pathway in Middle Africa shows a significant preference for the double mutant step S108N/N51I over S108N/C59R and that the evolution is at an earlier stage in this region than would be expected from the worldwide distribution, i.e., evolution to the quadruple mutation has not occurred as frequently as would be expected.

The distribution of mutations in South America is markedly different from the worldwide distribution, with single mutation S108N, double mutation N51I,S108N, and triple mutation N51I,S108N,I164L all overrepresented, whilst triple mutation N51I,C59R,S108N and the quadruple mutation are underrepresented. This suggests this region has a significant preference for the mutations involved in the most likely inferred pathway in this region S108N/N51I/I164L and the evolution is following a significantly different trajectory than the worldwide distribution.

Finally, we analysed the distribution of mutations in Melanesia and found the double mutation C59R,S108N is significantly overrepresented, whilst N51I,C59R,S108N and the quadruple mutation, which were both absent from this region, were underrepresented. This suggests the evolution in this region is at a much earlier stage than would be expected from the worldwide data.

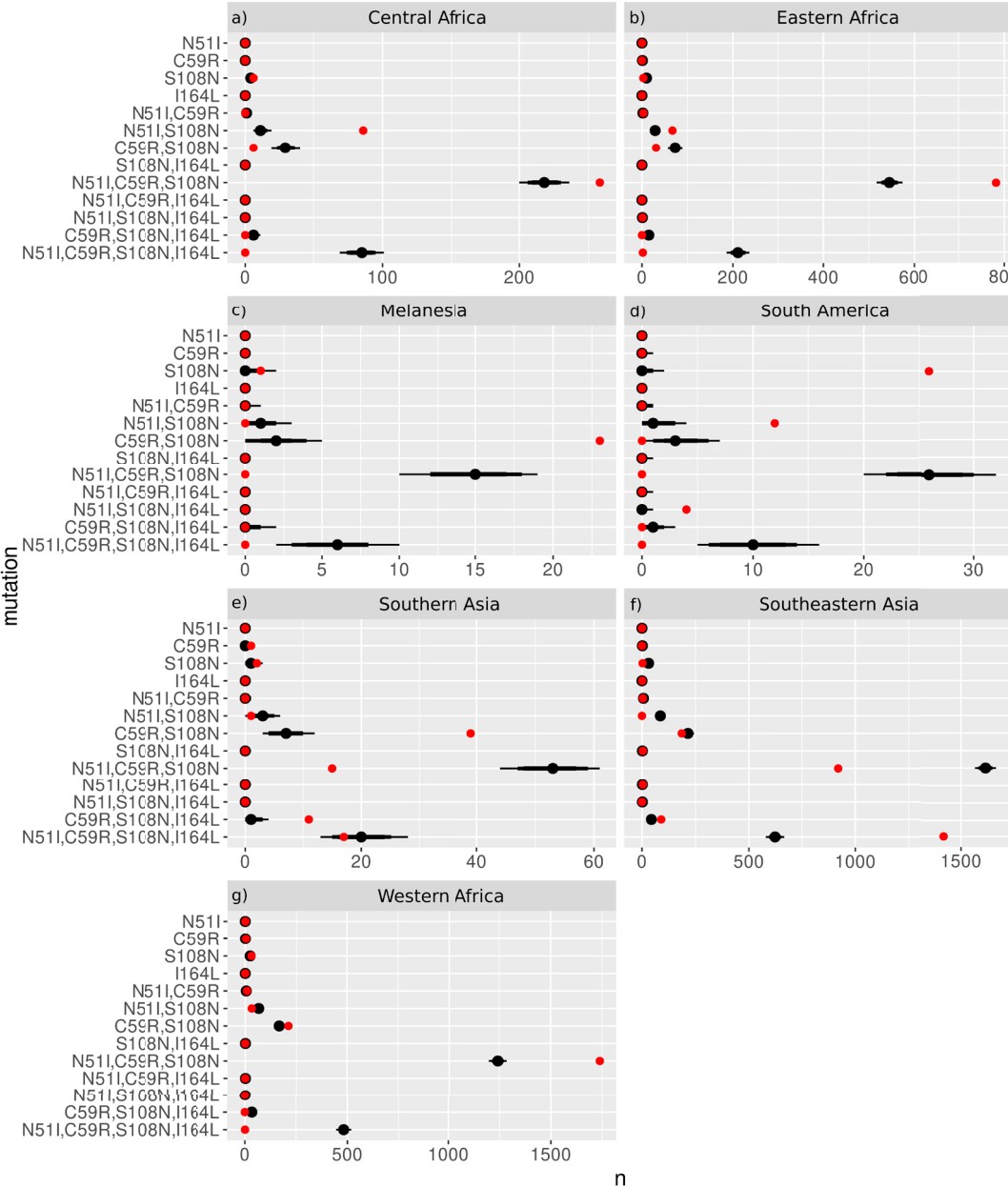

**Appendix 4—figure 1.** The frequency distributions of the *P. falciparum* dihydrofolate reductase (*Pf*DHFR) mutations from the 50,000 samples taken from the worldwide distribution with replacement for sample sizes (N) equal to the regional datasets from (**a**) Middle Africa (N=359), (**b**) Eastern Africa (N=904), (**c**) Melanesia (N=119), (**d**) South America (N=50), (**e**) Southern Asia (N=86), (**f**) Southeastern Asia (N=2650), and (**g**) Western Africa (N=2594). The red dots show the frequency of each mutation from the regional datasets and the black distributions show the 69%, 80%, and 90% quantile intervals of frequency distributions from the samples.

## *Pv*DHFR

The distribution of the four *Pv*DHFR mutations in South America is similar to the worldwide distribution (***Appendix 4—figure 2d***) with all of the observed mutations occurring at frequencies within or just outside the expected range. This supports our inference that the evolution in South America is following the same most likely pathway as the worldwide data.

In Eastern Africa, S117N was found to be overrepresented and S58R,S117N was found to be underrepresented compared to the worldwide distribution (***Appendix 4—figure 2a***). This suggests

evolution to the double mutation has not occurred as frequently in this region as would be expected from the worldwide distribution.

In Southern Asia, the distribution of mutations was as expected from the worldwide distribution, with the exception of N50I,S117N, which was overrepresented in this region (**Appendix 4—figure 2e**). This suggests the alternative pathway S117N/N50I inferred from the frequency data from this region is more prevalent than would be expected.

The frequency of the four *Pv*DHFR mutations in Eastern Asia, Southeastern Asia, and Melanesia is very low, therefore it is difficult to draw many conclusions about the frequency of mutations in these regions. From the distribution plots (**Appendix 4—figure 2b**, **Appendix 4—figure 2f** and **Appendix 4—figure 2c** for Eastern Asia, Southeastern Asia, and Melanesia, respectively), the mutations appear to be found at similar frequencies to what would be expected from the worldwide distribution, however due to the low frequencies it is difficult to conclude that definitively. More data is required from these areas to draw conclusions regarding the distribution of their mutations and the evolutionary pathways they appear to be following.

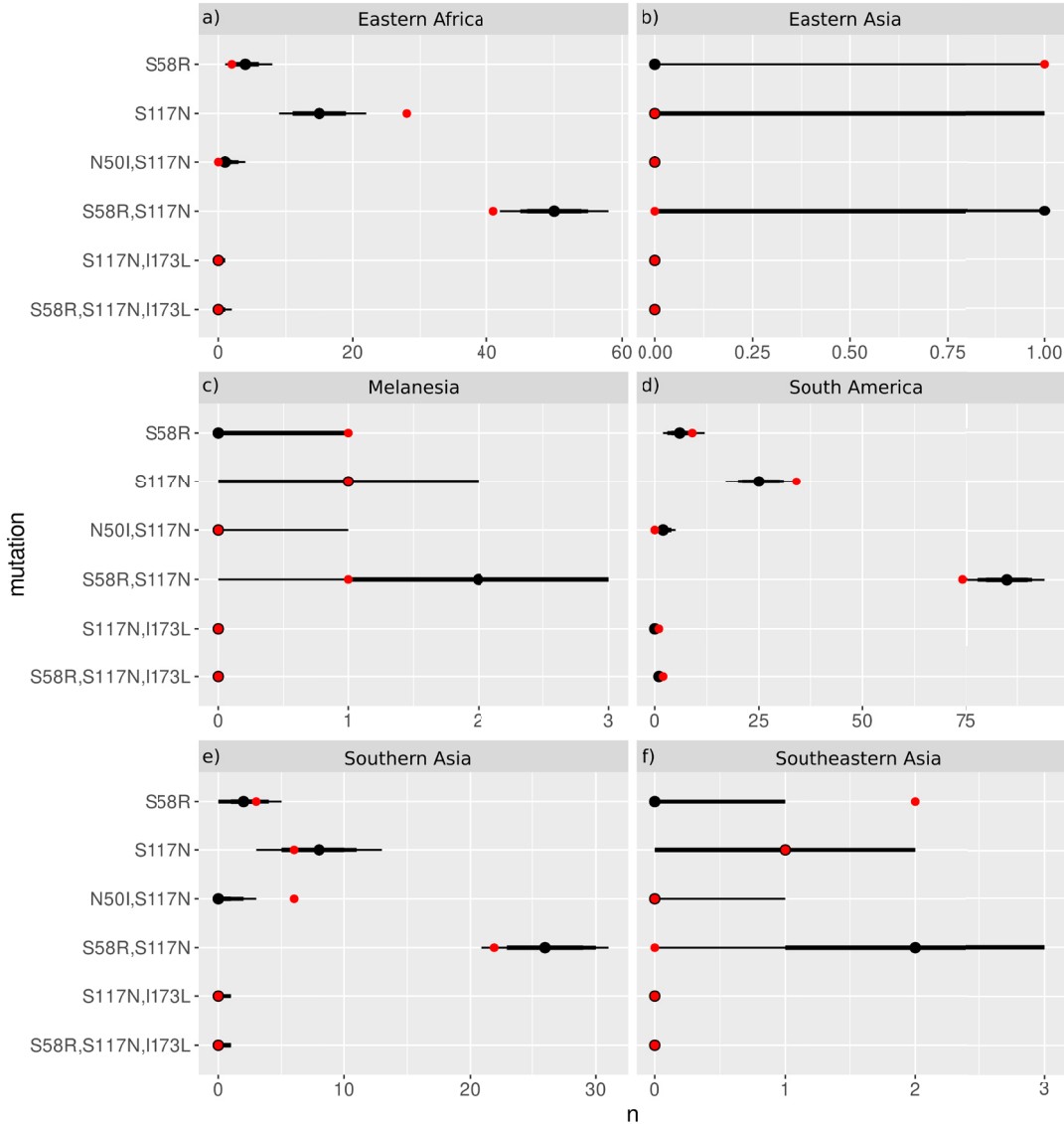

**Appendix 4—figure 2.** The frequency distributions of the *P. vivax* dihydrofolate reductase (*Pv*DHFR) mutations from the 50,000 samples taken from the worldwide distribution with replacement for sample sizes (N) equal to the regional datasets from (**a**) Eastern Africa (N=84), (**b**) Eastern Asia (N=12), (**c**) Melanesia (N=26), (**d**) South America

*Appendix 4—figure 2 continued*

(N=257), (**e**) Southern Asia (N=37), and (**f**) Southeastern Asia (N=334). The distribution from Central America was not analysed because it did not contain any combinations of the four *Pv*DHFR mutations being studied. The red dots show the frequency of each mutation from the regional datasets and the black distributions show the 69%, 80%, and 90% quantile intervals of frequency distributions from the samples.

