## [Editor Report]

Predicting the evolutionary path towards resistance through successive mutations is an important problem. This valuable study reports on the evolution of resistance to antifolates using computational predictions of changes in drug binding affinity. The findings generally rely on solid analyses although some of the claims are only partially supported because the computational predictions on the effects of mutations are only partially validated by prior experimental data. This study will be of interest to microbiologists interested in the evolution of drug resistance.

---

## [Decision Letter]

**Decision letter after peer review:**

Thank you for submitting your article "A computational method for predicting the most likely evolutionary trajectories in the stepwise accumulation of resistance mutations" for consideration by *eLife*. Your article has been reviewed by 3 peer reviewers, and the evaluation has been overseen by a Reviewing Editor and Christian Landry as the Senior Editor. The following individual involved in the review of your submission has agreed to reveal their identity: Adrian Serohijos (Reviewer #1).

Essential revisions:

Reviewer 1: It would be important to demonstrate how robust the results are given that the number of mutations and pathways being considered are limited. Reviewer 3 also mentions the lack of quantification of the performance of the method.

Reviewers 1 and 2 bring several points regarding the statistical analyses (p-values missing, model not converging, distribution of ddG values considered), the calculation relating fitness to the fraction of protein unbound by the drug, and the performance of Rosetta to estimate ddG. These elements would need to be corrected and the performance of Rosetta quantified and demonstrated.

Reviewer 2 suggested better dissecting the cases of epistasis to differentiate trivial cases due to the non-linearities inherent to the system, from other epistasis.

Reviewer 1 remarked that some evolutionary trajectories may not be independent of each other due to the possibility of gene flow.

Many relevant papers by Ogbunugafor and colleagues on studying epistasis and evolutionary trajectories on the same enzymes and in plasmodium were not considered here. It would seem important to cite this body of work and see how your results relate to their approaches and results. For example:

Ogbunugafor CB, Wylie CS, Diakite I, Weinreich DM, Hartl DL. Adaptive landscape by environment interactions dictates evolutionary dynamics in models of drug resistance. PLoS computational biology. 2016 Jan 25;12(1):e1004710.

Ogbunugafor CB, Hartl D. A pivot mutation impedes reverse evolution across an adaptive landscape for drug resistance in Plasmodium vivax. Malaria journal. 2016 Dec;15(1):1-0.

Ogbunugafor CB. The mutation effect reaction norm (mu‐rn) highlights environmentally dependent mutation effects and epistatic interactions. Evolution. 2022 Feb;76(S1):37-48.

Ogbunugafor CB, Eppstein MJ. Competition along trajectories governs adaptation rates towards antimicrobial resistance. Nature ecology and evolution. 2016 Nov 21;1(1):1-8.

*Reviewer #1 (Recommendations for the authors):*

1. There are five instances of incorrectly referenced figures throughout the text, "… (Figure ??) … ".

2. Appendix-figure 4 caption: "… gradient of the average…" Please clarify if this was a running average and over what interval window.

*Reviewer #2 (Recommendations for the authors):*

In this work the authors use a simple biophysical model to predict evolutionary trajectories of resistance to pyrimethamine – inhibitor of PfDHFR from *P. falciparum* and PvDHFR from *P. vivax* – pathogens causing malaria which presents a worldwide health concern. The authors use a simple fitness model that posits that selection coefficient -relative change in fitness between WT and mutant strains is determined by the fraction of unbound (to antibiotic inhibitor) DHFR. The population genetics simulations use the Kimura formula which is applicable to low mutation high selection regime where populations are monoclonal. The authors use computational tool Rosetta Flex ddG to assess binding of the antibiotic ligand to WT and mutant protein and compare their predicted evolutionary trajectories with lab evolution and data on naturally evolved variants worldwide and find semi-quantitative agreement, albeit sith significamt variation in detail.

The paper is of potential interest as it presents one of the first (but not the first) attempts to compare evolutionary dynamics based on biophysics inspired fitness model with laboratory evolution and natural data for very important problem of emergence and fixation of antibiotic resistant alleles. As such it can be a useful starting point for more detailed and biophysically realistic models of evolution of resistance against anti-DHFR drugs.

There are a number of issues – mostly technical but important – which limit potential impact and predictive power of this work. Let me list them in order of importance:

1) The fitness model whereby fitness of a variant is proportional to the fraction of free DHFR (and hence selection coefficient in Kimura formula is defined as relative difference in this quantity) is very simplistic. In fact earlier studies by Kaczer and Burns, Dean and Hartl and Rodrigues et al. (2016) cited here show that selection coefficient with respect to variation of DHFR is a more complex non-linear function of DHFR abundance, activity and – importantly – other enzymes in the folate metabolism pathway. Recent paper PMID: 26484862 established a proper fitness model for DHFR variation.

2) Rosetta ddG Flex does a mediocre job, to say the least in predicting binding free energy.

Table 1 shows that predictions in many cases are off quite substantially.

3) Equation 1 to predict fraction of unbound protein is not entirely accurate. The correct set of equations to determine this quantity is:

Lfree+LfreePfreeKd=L0

Pfree+LfreePfreeKd=P0

Lfree=L0+(Pfree−P0)

Pfree+(L0+(Pfree−P0))PfreeKd=P0

Pfree2Kd+Pfree(1+L0−P0Kd)−P0=0

Pfree=Kd2(−1−L0−P0Kd±(1+L0−P0Kd)+4P0K2)

=Kd2(−1−L0−P0Kd+1+2(L0+P0)Kd+(L0−P0Kd))

Where *P^free^* and *L^free^* are concentration of free (monomeric, unbound) protein and antibiotic Where ligand in solution and ^*P*0 and *L*0^ are their total concentrations. Under certain conditions this full result reduces to Equation 1 but it is important that the authors assess whether these conditions are indeed met in realistic cellular scenarios. In essence Equation 1 assumes that *L^free^*
^| *L*0^ but it is not clear whether this is a realistic condition.

4) Given that there are so many caveats and limitations to the underlying analysis the authors should revisit their results and discuss why some of the predictions are robust to these limitations and where the predictions fail due to the limitations of the analysis.

5) The authors point out to many cases when epistasis is observed but it can be a “trivial” epistasis due to the fact that *K_d_* that determines fraction of free proteins is related to '*G* in a non-linear way, see e.g. PMID: 21610162. It might be interesting to outline cases where epistasis is “trivial” and where it is related to effective interactions between mutation sites.

In summary this is an interesting work with some potential implications for predicting evolution of antibiotic resistance but technical concerns need to be addressed to make the foundation of the study more solid.

*Reviewer #3 (Recommendations for the authors):*

Major points and suggestions:

1. The comparison between the computational results and previous studies is done in a case-by-case manner. This is certainly informative. But it is somewhat dissatisfactory that a single summary metric (or a small number of them) is not provided as an overall performance measure of the computational method.

2. The presentation will benefit hugely if a fitness graph (based on hypercube representations, see for e.g. de Visser and Krug 2014, Figure 1c) is provided based on the free-energy change estimates from the computational method.

[Editors’ note: further revisions were suggested prior to acceptance, as described below.]

Thank you for resubmitting your work entitled "A computational method for predicting the most likely evolutionary trajectories in the stepwise accumulation of resistance mutations" for further consideration by *eLife*. Your revised article has been evaluated by Christian Landry (Senior Editor) and a Reviewing Editor.

The manuscript has been improved, but some remaining issues must be addressed, as outlined below. Following the submission of their respective reviews, we discussed with the reviewers, and the consensus is that the two significant points that follow remain problematic. Overall, the reviewers are excited about the development of computational tools to predict resistance. Still, at the same time, there remain major concerns about the actual data and model used to make predictions.

1) As raised by one of the reviewers in the initial review process, the RosettaFoldDDG poorly matches experimentally measured parameters. You now include a measure of correlation that confirms this poor accuracy. It, therefore, remains challenging to understand how the methods would work if the predicted measures do not reflect actual physical parameters. The reviewers would minimally require that the predictions of binding affinities are validated using alternative methods (computational or experimental).

2) One of the papers that are cited (2016 PNAS) to support the work actually shows that resistance, at least in bacteria, is very poorly predicted from the same parameters used here when considering a full fitness model. It would therefore be necessary that the authors use a fitness model to show that binding affinities are indeed predictive of resistance or that the values that are predicted using RosettaFoldDDG are indeed predictive.

*Reviewer #1 (Recommendations for the authors):*

The authors satisfactorily addressed my comments and concerns.

*Reviewer #2 (Recommendations for the authors):*

The authors addressed many editorial concerns raised by all reviewers.

I am still of the opinion that the premise of the study – to use biophysical model to explore evolutionary dynamics of resistant variants – is worthwhile and timely.

Nevertheless, my two most important essential concerns have not been addressed adequately making me question the technical validity of the study.

1) The authors completely misrepresent the key conclusion of the study of Rodrigues ate al 2016 PNAS. They suggest that Rodrigues et al. claimed that inhibitor binding to DHFR is the most important and predictive biophysical trait that determines fitness and more specifically IC50. In fact, the direct opposite is true – Fig, S8 of Rodrigues et al. showed that binding affinity to antibiotic is THE LEAST predictive of IC50 biophysical property and that there is no statistically significant correlation between K_i_ alone and IC50 for many variants and conditions explored in the 2016 Rodrigues at al PNAS. Essentially these authors showed that ignoring the effect of mutations on catalytic activity and proteins stability/abundance completely eliminates the predictive power of the biophysical model. This is the key conclusion of the 2016 paper which is completely misinterpreted in the present work. The authors of course may disagree with the conclusions of the 2016 study but in this case, they MUST present evidence of the validity of their simplified, naive biophysical model.

2) Table 1 still shows that there is no relationship between ddG predictions and reality and the authors themselves admit that the correlation is not statistically significant. Their justification of using Rosetta that there is no other method that provides prediction of binding affinities is both incorrect and disingenuous. Using such inaccurate predictions of the only parameter in their fitness model makes the approach devoid of predictive power.

---

## [Author Response]

Essential revisions:Reviewer 1: It would be important to demonstrate how robust the results are given that the number of mutations and pathways being considered are limited. Reviewer 3 also mentions the lack of quantification of the performance of the method.Reviewers 1 and 2 bring several points regarding the statistical analyses (p-values missing, model not converging, distribution of ddG values considered), the calculation relating fitness to the fraction of protein unbound by the drug, and the performance of Rosetta to estimate ddG. These elements would need to be corrected and the performance of Rosetta quantified and demonstrated.Reviewer 2 suggested better dissecting the cases of epistasis to differentiate trivial cases due to the non-linearities inherent to the system, from other epistasis.Reviewer 1 remarked that some evolutionary trajectories may not be independent of each other due to the possibility of gene flow.Many relevant papers by Ogbunugafor and colleagues on studying epistasis and evolutionary trajectories on the same enzymes and in plasmodium were not considered here. It would seem important to cite this body of work and see how your results relate to their approaches and results.

Thank you for bringing these papers to our attention, it is an interesting and important body of work.

For example:Ogbunugafor CB, Wylie CS, Diakite I, Weinreich DM, Hartl DL. Adaptive landscape by environment interactions dictates evolutionary dynamics in models of drug resistance. PLoS computational biology. 2016 Jan 25;12(1):e1004710.

We have cited this paper and briefly discussed the main results and how they can inform the analysis of our results (lines 634-651):

“Similarly, Ogbunugafor et al. (2016) demonstrated the importance of environmental variables on the evolution of drug resistance in PfDHFR by simulating evolution considering empirical growth measures and IC50 values across drug concentrations. They found that the rank order of allele fitness is a function of drug concentration and the preferred pathways to the fittest allele depends upon the environment. Therefore, whilst the model presented here captures the most likely pathways to resistance for high drug concentration, it is unable to predict important evolutionary dynamics that arise as a function of the environment. Therefore, the pathways predicted here should be considered the most likely pathways under sustained use of high concentrations of pyrimethamine.

Several investigations have also studied reverse evolution in PfDHFR (Ogbunugafor, 2022) and PvDHFR (Ogbunugafor and Hartl, 2016) by simulating evolution from the resistant to susceptible phenotype for different pyrimethamine concentrations, including the drugless environment. These studies also used measurements of growth rates and IC50 at the different drug concentrations in their simulations. They found S108N in PfDHFR and S117N in PvDHFR act as pivot point mutations and prevent reverse evolution to the susceptible phenotype. This has important consequences for AMR management strategies that aim to reduce resistance by the cessation of drug use.

Unfortunately, our model cannot be used to study reverse evolution because the fitness function is unable to quantify fitness in a drug-free environment.”

Ogbunugafor CB, Hartl D. A pivot mutation impedes reverse evolution across an adaptive landscape for drug resistance in Plasmodium vivax. Malaria journal. 2016 Dec;15(1):1-0.

We have cited this paper and discussed the results as described above and between lines 138-142:

“The epistasis between the four PfDHFR mutations and the four PvDHFR mutations may prevent reverse evolution from resistant to susceptible phenotypes, with S108N and S117N acting as pivot point mutations creating an epistatic ratchet against reverse evolution towards the wild-type and suggesting that the removal of pyrimethamine from use will not result in a reduction of resistance (Ogbunugafor et al. 2016; Ogbunugafor and Hartl 2016; Ogbunugafor 2022).”

Ogbunugafor CB. The mutation effect reaction norm (mu‐rn) highlights environmentally dependent mutation effects and epistatic interactions. Evolution. 2022 Feb;76(S1):37-48.

See above response.

Ogbunugafor CB, Eppstein MJ. Competition along trajectories governs adaptation rates towards antimicrobial resistance. Nature ecology and evolution. 2016 Nov 21;1(1):1-8.

We have cited this paper and discussed the results as described above and between lines 663-666:

“Interestingly, Ogbunugafor and Eppstein (2017) simulated the competition between successive alleles along adaptive trajectories in the evolution of pyrimethamine resistance in both PfDHFR and PvDHFR, and found that greedy trajectories are not always the most likely to be followed, with less greedy trajectories sometimes able to reach higher fitness more quickly.”

Reviewer #1 (Recommendations for the authors):1. There are five instances of incorrectly referenced figures throughout the text, "… (Figure ??) … ".

Thank you for highlighting this, all instances have been corrected.

2. Appendix-figure 4 caption: "… gradient of the average…" Please clarify if this was a running average and over what interval window.

We have clarified that this was a running average over intervals of 10 runs and has been stated in the caption.

Reviewer #2 (Recommendations for the authors):In this work the authors use a simple biophysical model to predict evolutionary trajectories of resistance to pyrimethamine – inhibitor of PfDHFR from *P. falciparum* and PvDHFR from P. vivax – pathogens causing malaria which presents a worldwide health concern. The authors use a simple fitness model that posits that selection coefficient -relative change in fitness between WT and mutant strains is determined by the fraction of unbound (to antibiotic inhibitor) DHFR. The population genetics simulations use the Kimura formula which is applicable to low mutation high selection regime where populations are monoclonal. The authors use computational tool Rosetta Flex ddG to assess binding of the antibiotic ligand to WT and mutant protein and compare their predicted evolutionary trajectories with lab evolution and data on naturally evolved variants worldwide and find semi-quantitative agreement, albeit sith significamt variation in detail.The paper is of potential interest as it presents one of the first (but not the first) attempts to compare evolutionary dynamics based on biophysics inspired fitness model with laboratory evolution and natural data for very important problem of emergence and fixation of antibiotic resistant alleles. As such it can be a useful starting point for more detailed and biophysically realistic models of evolution of resistance against anti-DHFR drugs.There are a number of issues – mostly technical but important – which limit potential impact and predictive power of this work. Let me list them in order of importance:1) The fitness model whereby fitness of a variant is proportional to the fraction of free DHFR (and hence selection coefficient in Kimura formula is defined as relative difference in this quantity) is very simplistic. In fact earlier studies by Kaczer and Burns, Dean and Hartl and Rodrigues et al. (2016) cited here show that selection coefficient with respect to variation of DHFR is a more complex non-linear function of DHFR abundance, activity and – importantly – other enzymes in the folate metabolism pathway. Recent paper PMID: 26484862 established a proper fitness model for DHFR variation.

We agree with the Reviewer that the real fitness landscape of DHFR is much more complex than our simple model, a fact we mention in the manuscript. However, one of the main aims of the project was to determine how well a simple model that could be easy to implement by many researchers, is able to capture real evolutionary trajectories. Rodrigues et al. 2016 show that binding affinity is the major determining factor in resistance, especially at later times in evolution, therefore the choice of only considering variations in binding affinity to see how well such a simple model can capture evolution is valid. Furthermore, the aim of the work was to determine a method to predict evolution that can be done purely using computational tools, without also including the use of wet lab experiments. There are many tools for the computational prediction of ligand binding affinity change upon mutation which only require a protein structure in complex with a ligand and can be used to quickly produce predictions of the impact of different mutations upon ligand binding. Of those methods, Rosetta Flex ddG is the state-of-the-art and has been shown to perform as well as molecular dynamics simulations at predicting binding affinity change (See Aldeghi et al., 2018 and 2019). Whilst characterising the entire fitness landscape would provide the most accurate model, it is not possible to computationally predict the other important factors, such as IC50 and catalytic activity, without the use of a wet-lab, which would defeat the purpose of the project. The remarkable agreement between the evolutionary trajectories predicted using our simple model, parameterised using only Rosetta Flex ddG predictions, suggests that this is an efficient and effective method that could be used by any researchers with access to a high-performance computing cluster and without the need for costly and time-consuming wet-lab experiments.

To avoid confusion, we have added further explanation of the motivations of our work in the Results section between lines 295-302:

“Whilst the fitness landscape of DHFR is a function of many factors including stability, abundance and activity (Rodrigues et al., 2016; Bershtein et al., 2015). Rodrigues et al., (2016) demonstrated that binding affinity is a major determinant of fitness, especially later in evolution. Therefore, we used a simple model that considered only predicted changes in binding free energy that would be easy to implement, undertaken purely computationally with good accuracy and without the need for extensive wet lab experiments. We sought to determine if simulated trajectories using this simple fitness model could capture observed evolutionary trajectories, despite not considering all factors that contribute to DHFR fitness.”

and in Methods section between lines 841-850.

“This equation is derived under the assumption that the concentration of unbound ligand *L_free_* ≈*L*_0_, where *L*_0_ is the total concentration. This captures the ideal scenario in which the concentration of the antimalarial drug is very high. We also assume that drug binding affinity is the most important determinant of protein fitness and ignore other factors such as IC50 and growth rate. We make this assumption because drug binding affinity was found to be a major determinant of fitness by Rodrigues et al., (2016). Our aim is to predict evolutionary trajectories without the need for experimental measurements and ligand binding prediction tools are readily available and easy to use, whereas it is more difficult to predict IC50 or growth rates. We therefore chose to use a simplified model of the system to approximate its behaviour, capture its general properties and make the model more tractable.”

2) Rosetta ddG Flex does a mediocre job, to say the least in predicting binding free energy.Table 1 shows that predictions in many cases are off quite substantially.

We agree with the Reviewer that the absolute predictions are not perfect, however no computational prediction tool is perfect and Flex ddG is the best available tool at this moment in time. However, and most importantly, Flex ddG is able to capture the general trend in the data. Furthermore, for the evolutionary simulations, we sample from the predicted Flex ddG distributions for each mutation, instead of just taking the average of the distributions,

3) Equation 1 to predict fraction of unbound protein is not entirely accurate. The correct set of equations to determine this quantity is:Lfree+LfreePfreeKd=L0Pfree+LfreePfreeKd=P0Lfree=L0+(Pfree−P0)Pfree+(L0+(Pfree−P0))PfreeKd=P0Pfree2Kd+Pfree(1+L0−P0Kd)−P0=0
Pfree=Kd2(−1−L0−P0Kd±(1+L0−P0Kd)+4P0K2)=Kd2(−1−L0−P0Kd+1+2(L0+P0)Kd+(L0−P0Kd))

Where *P*^*free*^ and *L*^*free*^ are concentration of free (monomeric, unbound) protein and antibiotic Where ligand in solution and ^P0 and L0^ are their total concentrations. Under certain conditions this full result reduces to Equation 1 but it is important that the authors assess whether these conditions are indeed met in realistic cellular scenarios. In essence Equation 1 assumes that L^free | L0^ but it is not clear whether this is a realistic condition.

The Reviewer is correct regarding the assumption that *L_free_* ≈*L*_0_ in the model. The model represents the ideal scenario in which there is a high concentration of drug in the cell which enables us to approximate the fitness of each variant by just considering binding affinity (of course under the simplifying assumption that binding affinity is the only determinant of fitness and ignoring ither factors such as IC50, as discussed above). Whilst we are aware this may not be the case in a real cell, idealised models are used throughout science to approximate the behaviour of a system, capture general properties and make modelling complex systems more tractable. Indeed, we do not claim this simple model is able to capture all aspects of the evolution of drug resistance, but the aim of the project was to determine how well such a simple model, parameterised using computational predictions, could capture observed evolutionary trajectories. We would like to thank the Reviewer for highlighting that we have not made this clear in the text and have added an explanation between lines 841-850:

“This equation is derived under the assumption that the concentration of unbound ligand *L_free_* ≈*L*_0_, where *L*_0_ is the total concentration. This captures the ideal scenario in which the concentration of the antimalarial drug is very high. We also assume that drug binding affinity is the most important determinant of protein fitness and ignore other factors such as IC50 and growth rate. We make this assumption because drug binding affinity was found to be a major determinant of fitness by Rodrigues et al., (2016). Our aim is to predict evolutionary trajectories without the need for experimental measurements and ligand binding prediction tools are readily available and easy to use, whereas it is more difficult to predict IC50 or growth rates. We therefore chose to use a simplified model of the system to approximate its behaviour, capture its general properties and make the model more tractable.”

4) Given that there are so many caveats and limitations to the underlying analysis the authors should revisit their results and discuss why some of the predictions are robust to these limitations and where the predictions fail due to the limitations of the analysis.

We agree with the Reviewer that there are many caveats and limitations to this method. Though we did include a discussion of this in the original manuscript, we have now expanded our analysis and discussion of the limitations of the model.

a) Our model cannot account for environmental variables, which we discuss between lines 634-642 in the Discussion:

“Similarly, Ogbunugafor et al. (2016) demonstrated the importance of environmental variables on the evolution of drug resistance in PfDHFR by simulating evolution considering empirical growth measures and IC50 values across drug concentrations. They found that the rank order of allele fitnesses is a function of drug concentration and the preferred pathways to the fittest allele depends upon the environment. Therefore, whilst the model presented here captures the most likely pathways to resistance for high drug concentration, it is unable to predict important evolutionary dynamics that arise as a function of the environment. Therefore, the pathways predicted here should be considered the most likely pathways under sustained use of high concentrations of pyrimethamine.”

b) Our model cannot be used to simulate reverse evolution, which we mention in the Discussion between lines 643-651:

“Several investigations have also studied reverse evolution in PfDHFR (Ogbunugafor, 2022) and PvDHFR (Ogbunugafor and Hartl, 2016) by simulating evolution from the resistant phenotype to the susceptible phenotype for different pyrimethamine concentrations, including the drugless environment. These studies also used measurements of growth rates and IC50 at the different drug concentrations in their simulations. They found S108N in PfDHFR and S117N in PvDHFR act as pivot point mutations and prevent reverse evolution to the susceptible phenotype. This has important consequences for AMR management strategies that aim to reduce resistance by the cessation of drug use. Unfortunately, our model cannot be used to study reverse evolution because the fitness function is unable to quantify fitness in a drug-free environment.”

c) Our model does not account for other factors that determine DHFR fitness, such as IC50 and growth rate, which we mention in Results section between lines 295-302:

“Whilst the fitness landscape of DHFR is a function of many factors including stability, abundance and activity (Rodrigues et al., 2016; Bershtein et al., 2015). Rodrigues et al., (2016) demonstrated that binding affinity is a major determinant of fitness, especially later in evolution. Therefore, we used a simple model that considered only predicted changes in binding free energy that would be easy to implement, undertaken purely computationally with good accuracy and without the need for extensive wet lab experiments. We sought to determine if simulated trajectories using this simple fitness model could capture observed evolutionary trajectories, despite not considering all factors that contribute to DHFR fitness.”

d) We mention the fact that our model is only applicable in the weak mutation regime and discuss what this might mean for our results 660-678:

“A further limitation of the evolutionary model is that it operates in the weak mutation regime, in which the mutation rate is so low that mutations appear and fix in isolation. However, this assumption breaks down when considering large microbial populations where clonal interference means that mutations can arise simultaneously and compete for fixation (Gerrish and Lenski, 1998). This can lead to a process known as ‘greedy adaptation’, in which the mutation of larger beneficial effect is fixed with certainty (Jain et al., 2011). Interestingly, Ogbunugafor and Eppstein (2017) simulated the competition between successive alleles along adaptive trajectories in the evolution of pyrimethamine resistance in both PfDHFR and PvDHFR, and found that greedy trajectories are not always the most likely to be followed, with less greedy trajectories sometimes able to reach higher fitness more quickly.

Clonal interference has been shown to emerge rapidly in laboratory cultivated *P. falciparum*, where the parasite cycles through only asexual stages, suggesting it may influence the dynamics of the emergence of resistance (Jett et al., 2020). The evolutionary model used here may therefore overestimate the fixation probability of mutations with milder beneficial effects and underestimate the fixation probability of mutations with larger beneficial effects (de Visser and Krug 2014). Future iterations of this work could be improved by making using a fixation probability that models clonal interference such as the work of Gerrish and Lenski (1998) and Campos et al., (2004). However, such models are more difficult to implement as they can require species-specific derivations for certain functions.”

e) We discuss the fact that our model does not account for thermodynamic stability between lines 676-693:

“Mutations occurring at a drug-binding site may also reduce the protein’s thermodynamic stability (Wang et al., 2002) and therefore may not be selected for, even if they improve the resistance phenotype. However, our model does not take thermodynamic stability into account, and so may predict certain mutations to be beneficial to fitness even if they are actually deleterious because they decrease the protein's stability. This may lead to incorrect prediction of pathways. Our model may be improved by including selection for mutations that do not reduce thermodynamic stability relative to the wild-type enzyme, similar to the work by Rotem et al. (2018). However, it must also be noted that most proteins are marginally stable (Vogl et al., 1997; Ruvinov et al., 1997), a property which may have evolved either as an evolutionary spandrel (Taverna and Goldstein 2002; Goldstein, 2011) (a characteristic that arises as a result of non-adaptive processes which is then used for adaptive purposes (Gould et al., 1979) or due to selection for increased flexibility to improve certain functionalities (Zavodszky et al., 1998 and Tsou et al., 1998)). Therefore, the model would also need to account for the fact that a resistance mutation that increases protein stability relative to the wildtype stability may also result in a reduction in fitness.

In summary, the real fitness landscape of DHFR is much more complex than we can capture with our model, which only accounts for changes in binding affinity. Therefore, our model may predict evolutionary pathways to be likely when they are in fact inaccessible if a step in the pathway is deleterious to other aspects of the fitness landscape. The main advantage of our model is its simplicity, requiring only predictions of change in drug binding affinity to capture evolutionary pathways under the assumption of sustained high drug concentration.”

e) We also highlight the dependence of the model upon the accuracy of the Flex ddG predictions in the Discussion lines 694-703:

“Furthermore, our method uses Flex ddG to predict changes in drug binding free energy, which are then used to parameterise the fitness function of the evolutionary model. Therefore, the success of the method relies upon the accuracy of these predictions. As we established with the PfDHFR mutations, Flex ddG performs well for the single mutations but is less accurate for the higher order mutations, meaning pathways may be predicted to be inaccessible by our model but which in reality are accessible. However, in general Flex ddG performed well and was able to capture the general trend of the data for the mutations considered here, and so the model showed good agreement with experimentally determined pathways. Our model could also be parameterised using molecular dynamics, which may provide more accurate predictions of drug binding affinity, though with a much higher computational cost.”

5) The authors point out to many cases when epistasis is observed but it can be a “trivial” epistasis due to the fact that K_d_ that determines fraction of free proteins is related to 'G in a non-linear way, see e.g. PMID: 21610162. It might be interesting to outline cases where epistasis is “trivial” and where it is related to effective interactions between mutation sites.

Following the Reviewers advice, we have included a discussion on trivial vs non-trivial epistasis in the Introduction (lines 47-65):

“Epistasis can arise either as a result of direct or indirect interactions between residues resulting in non-additivity in a physical property, such as stability and affinity (non-trivial epistasis), or as a result of a non-linear mapping from sequence to protein function or fitness (trivial epistasis). Trivial epistasis, also known as nonspecific epistasis, effects a much larger number of mutations than nontrivial epistasis, because all mutations that impact a physical property that maps nonlinearly to function or fitness will interact epistaically with one another. Trivial epistasis has been widely studied and observed for mutations that are additive with respect to physical properties including stability, ligand binding affinity, function and folding (Gong et al.,2013; Bloom et al., 2010; McKeown et al., 2014; Lunzer et al., 2005).

Non-trivial (or specific) epistasis, impacts a smaller number of mutations as it requires direct or indirect interactions between amino acid side chains (Dickinson et al., 2013; Pantoliano et al., 1989; Olson et al., 2014) or ligand (Adams et al., 2019; Anderson et al. 2015) or a dependence of one mutation on a structural change caused by another (Ortlund et al., 2007; Dellus-Gur et al., 2015). Non-trivial epistasis is therefore associated with stronger evolutionary constraints, historical contingency and lower reversibility of substitutions in evolutionary trajectories. Here, we are explicitly considering mutations interacting via non-trivial epistasis, where their interactions result in non-additivity in drug binding affinity. However, our fitness function maps binding affinity to fitness nonlinearly, and so epistasis may also arise at the fitness level in our simulations of evolutionary trajectories, but we do not investigate this further.”

Reviewer #3 (Recommendations for the authors):Major points and suggestions:1. The comparison between the computational results and previous studies is done in a case-by-case manner. This is certainly informative. But it is somewhat dissatisfactory that a single summary metric (or a small number of them) is not provided as an overall performance measure of the computational method.

To summarise the overall performance of the computational method, we calculated the intersection between the top pathways predicted by our model and those predicted by Lozovosky et al., 2009, and the average overlap, calculated by taking the average of the intersect for all depths, thus weighting the top ranks higher.

The overall intersect between the top 10 PfDHFR pathways predicted by our model and the pathways predicted by Lozovosky et al., 2009 is 0.6 and the average overlap is 0.687.

The overall intersect between the top 4 PvDHFR pathways predicted by our model and the pathways predicted by Jiang et al. 2013 is 0.75 and the average overlap is 0.604.

These metrics of the overall performance of the model have been added to the Results section.

2. The presentation will benefit hugely if a fitness graph (based on hypercube representations, see for e.g. de Visser and Krug 2014, Figure 1c) is provided based on the free-energy change estimates from the computational method.

We thank the Reviewer for the suggestion, we have added a fitness hypercube figure to the Appendix.

[Editors’ note: what follows is the authors’ response to the second round of review.]

The manuscript has been improved, but some remaining issues must be addressed, as outlined below. Following the submission of their respective reviews, we discussed with the reviewers, and the consensus is that the two significant points that follow remain problematic. Overall, the reviewers are excited about the development of computational tools to predict resistance. Still, at the same time, there remain major concerns about the actual data and model used to make predictions.1) As raised by one of the reviewers in the initial review process, the RosettaFoldDDG poorly matches experimentally measured parameters. You now include a measure of correlation that confirms this poor accuracy. It, therefore, remains challenging to understand how the methods would work if the predicted measures do not reflect actual physical parameters. The reviewers would minimally require that the predictions of binding affinities are validated using alternative methods (computational or experimental).

We do not agree that the predictions poorly match experimental parameters. We observed a correlation of 0.611 between the Rosetta Flex ddG and experimental data with a p-value of 0.04. That is very similar to the correlation reported by Aldeghi et al. 2018 and 2019 (the observed maximum correlation of 0.67 depending on the scoring function used) when testing Flex ddG against experimental data and is comparable to the performance of molecular dynamics and mCSM-lig, all accepted and frequently-used methods for computationally predicting ligand binding affinity. Of course, the predictions could be improved, however these are the tools that are currently the best available.

2) One of the papers that are cited (2016 PNAS) to support the work actually shows that resistance, at least in bacteria, is very poorly predicted from the same parameters used here when considering a full fitness model. It would therefore be necessary that the authors use a fitness model to show that binding affinities are indeed predictive of resistance or that the values that are predicted using RosettaFoldDDG are indeed predictive.

We apologise if it had appeared that we had misunderstood the paper. We had concentrated on the final Results section of that paper that states “changes in the ability to escape drug (Ki) are they key determinants of drug resistance, especially at later stages of evolution (double and triple mutants), because variations in protein abundance are comparably smaller.” Therefore, while the main message of the paper may be that the best predictions are achieved using a full fitness model, the suggestion is that when protein abundance variation is low, then binding affinity becomes the most important parameter. We have clarified this in our paper. Furthermore, we do not claim that binding affinities on their own are fully predictive of resistance and we mention several times in the paper that resistance is a result of a trade-off between many parameters, however despite this we are able to achieve good agreement with experimentally determined stepwise evolutionary pathways to resistance when only considering binding affinity. Whilst we agree that a full fitness model would provide better predictions than our model, the purpose of our work is to see how well we can predict evolution using existing computational tools to parameterise the fitness model (i.e. creating a fully computational predictive model without the need for any wet lab experiments). As far as we are aware, it is not yet possible to computationally predict the other parameters required for the full fitness model, whereas prediction tools are available for drug binding affinity. The agreement we observe despite the simplicity of our model suggests that, in some cases at least, determining changes in binding affinity may be sufficient to predict at least the most likely evolutionary pathways to resistance.

Reviewer #2 (Recommendations for the authors):The authors addressed many editorial concerns raised by all reviewers.I am still of the opinion that the premise of the study – to use biophysical model to explore evolutionary dynamics of resistant variants – is worthwhile and timely.

We would like to thank the reviewer for acknowledging that the premise of our work is worthwhile and timely.

Nevertheless, my two most important essential concerns have not been addressed adequately making me question the technical validity of the study.1) The authors completely misrepresent the key conclusion of the study of Rodrigues ate al 2016 PNAS. They suggest that Rodrigues et al. claimed that inhibitor binding to DHFR is the most important and predictive biophysical trait that determines fitness and more specifically IC50. In fact, the direct opposite is true – Fig, S8 of Rodrigues et al. showed that binding affinity to antibiotic is THE LEAST predictive of IC50 biophysical property and that there is no statistically significant correlation between K_i_ alone and IC50 for many variants and conditions explored in the 2016 Rodrigues at al PNAS. Essentially these authors showed that ignoring the effect of mutations on catalytic activity and proteins stability/abundance completely eliminates the predictive power of the biophysical model. This is the key conclusion of the 2016 paper which is completely misinterpreted in the present work. The authors of course may disagree with the conclusions of the 2016 study but in this case, they MUST present evidence of the validity of their simplified, naive biophysical model.

We apologise if it appeared that we had misinterpreted the main conclusions of the 2016 Rodrigues PNAS paper, we were focussing on the last Results section of the paper in which it states K_i_ is the most important factor, especially at later stages of evolution when variations in protein abundance are small. We agree that accounting for all of the biophysical properties would produce a better model, and we state this in the manuscript. However, we chose to use a model that only considers changes in binding affinity because there are existing computational methods to predict these values, whereas as far as we are aware, there are not any purely computational methods to predict the other biophysical properties. Of course, if more computational tools become available, we would incorporate as many biophysical properties as possible into the model, but for the method to be purely computational, we can only use predictions of binding affinity at this time. The principal purpose of our work is to see how well we can predict stepwise evolutionary pathways to resistance by using computational methods only, and so create a predictive method that is efficient and accessible. We would argue that the fact that pathways to resistance predicted by our simple model that only uses predictions of binding affinity agrees well with experimentally determined evolutionary pathways is evidence of the validity of using our simplified biophysical model. We have expanded on this and have included further discussion in our paper of the limitations of our model as well as highlighting that there are situations in which it may not be applicable.

2) Table 1 still shows that there is no relationship between ddG predictions and reality and the authors themselves admit that the correlation is not statistically significant. Their justification of using Rosetta that there is no other method that provides prediction of binding affinities is both incorrect and disingenuous. Using such inaccurate predictions of the only parameter in their fitness model makes the approach devoid of predictive power.

We completely disagree with the reviewer’s comments and it is incorrect to say that we admit the correlation is not statistically significant. As we report in the Results, the correlation between ddG predictions and the experimental data is 0.611, which is comparable to correlations in the literature (Aldeghi et al. 2018 and 2019, Pires et al. 2016) and it is statistically significant with a p-value of 0.04, which is significant in the 95% confidence interval.

We are confused why the Reviewer says it is ‘incorrect and disingenuous’ to claim there are no other methods to predict binding affinity. To clarify, we mean computational predictions of binding affinity, and as far as we are aware the currently available methods for predicting ligand binding affinity changes upon mutation are the machine learning tool mCSM-lig, Rosetta Flex ddG and molecular dynamics, which we mention in the manuscript. Rosetta Flex ddG performs comparably to all the other computational methods. We would be interested to know of the other available tools the Reviewer is referring to as they would be useful to our work. We also disagree our approach is ‘devoid of predictive power’ since our predicted pathways agree well with experimentally determined evolutionary pathways to resistance.